# Human-type sialic acid receptors contribute to avian influenza A virus binding and entry by hetero-multivalent interactions

Mengying Liu[1], Liane Z. X. Huang[1], Anthony A. Smits[1], Christian Büll[2,3], Yoshiki Narimatsu [2], Frank J. M. van Kuppeveld [1], Henrik Clausen [2], Cornelis A. M. de Haan [1✉] & Erik de Vries [1✉]

Establishment of zoonotic viruses, causing pandemics like the Spanish flu and Covid-19, requires adaptation to human receptors. Pandemic influenza A viruses (IAV) that crossed the avian-human species barrier switched from binding avian-type α2-3-linked sialic acid (2-3Sia) to human-type 2-6Sia receptors. Here, we show that this specificity switch is however less dichotomous as generally assumed. Binding and entry specificity were compared using mixed synthetic glycan gradients of 2-3Sia and 2-6Sia and by employing a genetically remodeled Sia repertoire on the surface of a Sia-free cell line and on a sialoglycoprotein secreted from these cells. Expression of a range of (mixed) 2-3Sia and 2-6Sia densities shows that non-binding human-type receptors efficiently enhanced avian IAV binding and entry provided the presence of a low density of high affinity avian-type receptors, and vice versa. Considering the heterogeneity of sialoglycan receptors encountered in vivo, hetero-multivalent binding is physiologically relevant and will impact evolutionary pathways leading to host adaptation.

[1] Virology group, Division of Infectious Diseases and Immunology, Department of Biomolecular Health Sciences, Faculty of Veterinary Medicine, Utrecht University, Utrecht, The Netherlands. [2] Center for Glycomics, Departments of Cellular and Molecular Medicine and Odontology, Faculty of Health Sciences, University of Copenhagen, Blegdamsvej 3, Copenhagen, Denmark. [3] Department of Biomolecular Chemistry, Institute for Molecules and Materials, Radboud University, Nijmegen, The Netherlands. ✉email: c.a.m.dehaan@uu.nl; e.devries@uu.nl

Human influenza A virus (IAV) strains originate from an animal reservoir by zoonotic transfer. Waterfowl are the natural IAV reservoir while poultry and swine function as an intermediate reservoir[1]. To acquire transmissibility within the human population, avian IAVs need to overcome differences between the human and avian sialic acid (Sia) receptor repertoire[2,3]. Sias, which in avians and humans exclusively occur as the N-acetylneuraminic acid type, are monosaccharides terminating the glycan chains attached to membrane-associated and secreted glycoproteins and glycolipids. IAV infection by genotypes H1 to H16 is initiated by binding of hemagglutinin (HA) on the viral envelope to epithelial cell surface-exposed Sia receptors. A paradigm in IAV biology is the dichotomy in receptor specificity. Avian IAVs bind to a Sia linked by an α2-3-linkage to the penultimate galactose and human IAVs bind to α2-6-linked Sias[4,5]. Binding specificity matches the dominant Sia-linkage type at the preferred infection sites, being the enteric tract of waterfowl and upper respiratory tract of humans[6–8]. Often, α2-3 and α2-6 Sia linkages are found coinciding, at various ratios, on the different types of glycoproteins and glycolipids found at these sites which is thought to support the occasional infection of specific stretches of the swine or human respiratory tract by some avian IAVs. A positive correlation between receptor binding avidity and infection efficiency has been demonstrated[9,10], but ample examples suggest the lack of a strict correlation[8,11–14]. This controversy likely originates from large differences in complexity of the artificial, rather homogeneous, receptor surfaces that are employed for determining IAV binding avidity and specificity and the highly diverse sialoglycan repertoire presented on cultured cells and respiratory epithelium. N- and O-linked glycans attached to proteins and glycolipids each encompass a specific range of glycan structures that vary by linkages and extent of sialylation and differ in diversity and distribution between species, tissue and cell type. This is important as receptor binding avidity of IAVs is, apart from the α2-3/α2-6 linkage-type, affected by the underlying structure of the glycan[12,15]. For instance, H3N2 causing seasonal human flu has evolved since 1968 to a binding preference for multi-antennary glycans extended by multiple LacNAc repeats[12,16].

Individual HA-receptor interactions are short-lived because of their low affinity ($K_D$ ~0,5 to 20 mM)[17–21]. This leads to a relatively low initial virus binding rate, while high-avidity virus binding results from multivalent interaction with a receptor-coated surface[22–25] thereby promoting efficient virus entry. Sialoglycans that are identified as high-avidity IAV binders by in vitro binding assays, for example glycan arrays[13,26,27], are considered prime candidates for supporting IAV binding and entry. However, epithelial cell surfaces display a highly heterogeneous Sia repertoire of low and high affinity receptors that may simultaneously interact with a virus particle. Such a binding mode has been described for glycan binding by lectins, glycosidases and glycosyltransferases and is referred to as hetero-multivalent binding[28–31]. Importantly, it has never been investigated to which extent low affinity interactions contribute to high-avidity virus binding and entry by such a hetero-multivalent binding mode.

Here, we challenge the paradigm of the canonical 2-3/2-6Sia receptor dichotomy. We show that, by hetero-multivalent binding, synthetic sialoglycans which by themselves cannot support binding can compensate for insufficient amounts of preferred receptors for avian as well as human IAV strains. Subsequently, we systematically tune the sialoglycan composition on the cell membrane of a Sia-deficient HEK293 cell line (HEK$^{\Delta Sia}$)[32] by co-transfection with specific sialyltransferases. In this way mixed 2-3/2-6Sia gradients are presented on the cell surface, allowing virus entry studies, and on a sialoglycoprotein secreted from these cells, allowing to study the binding dynamics by biolayer interferometry (BLI). The results show that Sia receptors hitherto

not supposed to support IAV binding, actually contributed to enhanced binding and infection.

## Results

**Hetero-multivalent IAV binding to synthetic glycans**. To assess the contribution of low-affinity interactions to hetero-multivalent IAV binding to heterogenous receptor surfaces, we determined the binding of human seasonal H3N2 strain VI75 and avian H5N1 strain HU02 to BLI sensors coated with (mixtures of) 2-6- and 2-3-linked Sias. These viruses display the canonical Sia linkage-type binding specificity for human-type [2-6S(LN)2] or avian-type [2-3 S(LN)2] receptors respectively (Fig. 1a, b). At 7.5% to 15% density of these specific receptors only very limited virus binding was observed (Fig. 1c–f; black lines). However, by complementing a low density of the high-affinity receptor with the apparently non-binding low-affinity receptor (e.g., 7.5% 2-3 S(LN)2 plus 92.5% 2-6(LN)2 for HU02 in Fig. 1d), the binding rate of HU02 or VI75 could be strongly enhanced. We conclude that Sias which are generally considered to be non-binding receptors (2-3Sia for human IAVs or 2–6Sia for avian IAVs), can efficiently support binding by hetero-multivalency effects when the high-affinity Sia receptors are present at a low density that by itself cannot support binding.

Next, the IAV binding rate enhancement by low-affinity receptors was quantified over the full density range of the high-affinity receptor complemented by either the low-affinity receptor or the asialoglycan (LN)2 (Fig. 2). Virus binding rates were determined from the slope of the binding curves (examples shown in Supplementary Fig. 1a, b). Binding rates were normalized to the highest binding rate for each virus and plotted against the relative density of the high-affinity receptor (Fig. 2a). Four human H3N2 strains isolated between 1975 and 2002 (VI75, BK79, WU95, FU02) displayed positive hetero-multivalent binding effects (Fig. 2a–d) as complementation of the high-affinity 2-6S(LN)2 receptors with low-affinity 2-3S(LN)2 receptors (magenta lines) resulted in a leftward-shifted binding curve in comparison to complementation by asialoglycans (black lines). The lab-adapted, human-derived, H1N1 strain PR8 isolated in 1934 (Fig. 2e) and the avian H5N1 strain HU02 (Fig. 2f) displayed a similar shift when the high-affinity 2-3S(LN)2 receptors where complemented with the low-affinity 2-6 S(LN)2 receptors. To confirm that hetero-multivalent binding can have a positive effect on binding of other avian IAVs we show similar effects on an H5N8 virus strain and on another H5N1 IAV (Supplementary Fig. 2). We conclude that positive hetero-multivalent binding effects by low-affinity receptors are widespread, occurring for human viruses specifically binding 2-6Sia receptors as well as avian viruses specifically binding 2-3Sia receptors. At low densities of the high-affinity receptor, surface binding is absolutely dependent on hetero-multivalent binding effects. As shown for VI75 (Supplementary Fig. 3), the magnitude of a positive hetero-multivalency effect also depends on the density of the low-affinity receptor. The effect of the low-affinity receptor density is more gradual than of the high-affinity receptor density (note the smaller slope of the lines of Supplementary Fig. 3 in comparison to Fig. 2).

**Genetic tuning of the sialoglycan content in HEK$^{\Delta Sia}$ cells**. The maximal synthetic sialoglycan receptor density of ~0.0036 Sia moieties per nm$^2$ [23] on a BLI biosensor surface is low in comparison to ~1 to 6 Sias per nm$^2$ in the cell glycocalyx[33]. Also, glycan presentation on a flat biosensor surface poorly mimics the thick glycocalyx in which a large part of the Sias will be shielded from interaction with virus particles. To mimic in vivo Sia density and presentation more closely an experimental setup (Fig. 3a) was established that enables direct comparison of virus binding and

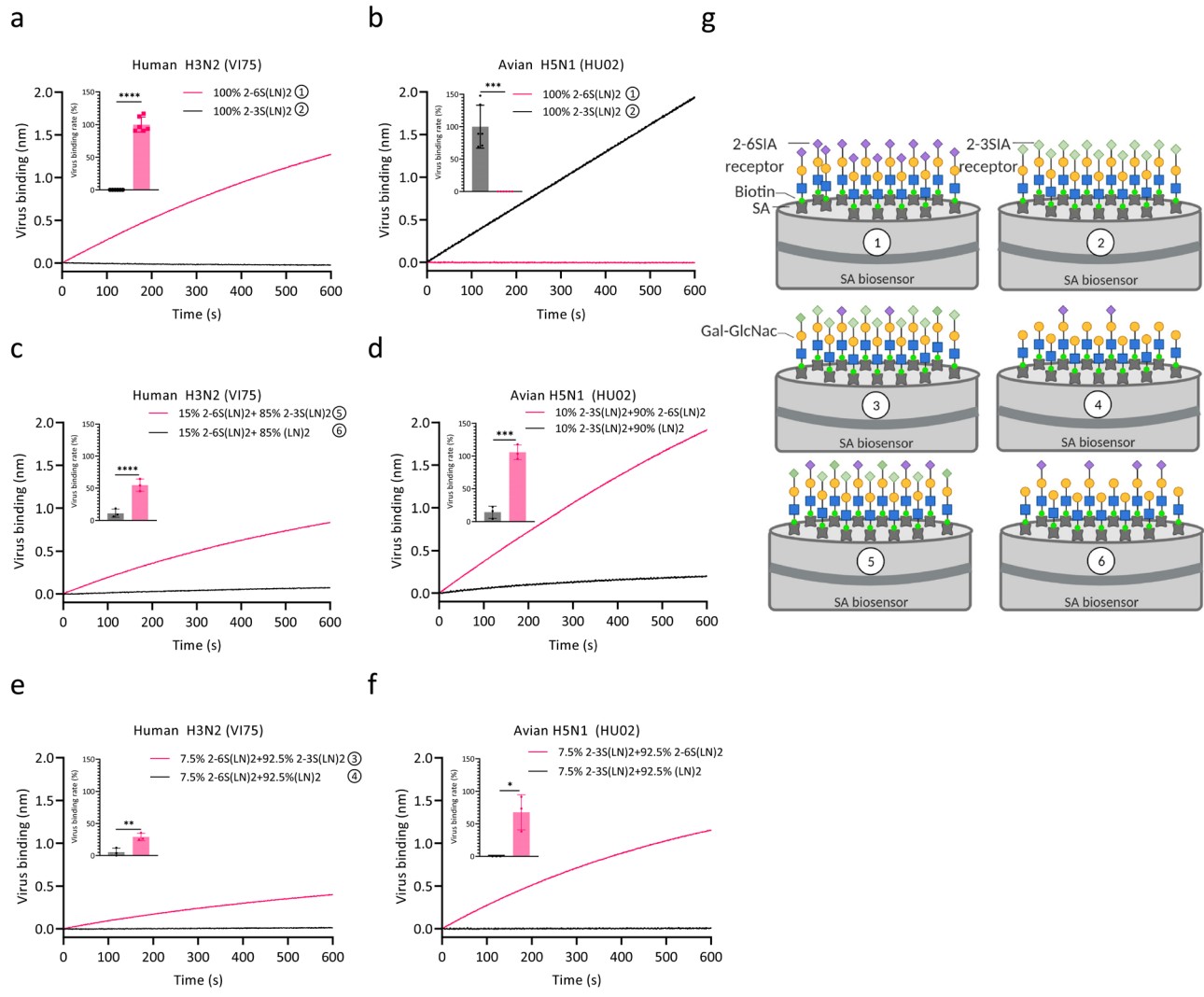

**Fig. 1 Low-affinity receptors enhance IAV binding rate in presence of a low density of high-affinity receptors. a–f**, ~100 pM suspensions of human H3N2 strain VI75 (**a**, **c**, **e**) or avian H5N1 strain HU02 (**b**, **d**, **f**) were associated for 600 s to BLI sensors loaded to maximum density (0.6 pM/cm²) with 100% 2-6 S(LN)2 (Neu5Aca2-6Galβ1-4GlcNAcβ1-3Galβ1-4GlcNAc) or 2-3 S(LN)2 (**a**, **b**) or with mixtures of these two synthetic glycans and the asialo glycan LN2 (Galβ1-4GlcNAcβ1-3Galβ1-4GlcNAc) at relative amounts as indicated in the figure (**c–f**) and illustrated in (**g**) as an example for panels **a**, **c** and **e** (2-6 Sia, purple diamonds; 2-3 Sia, light green diamonds; galactose, yellow; N-acetylglucosamine, blue).The inserts display the binding rate relative to the maximal binding rate obtained to sensors loaded with 100% 2-6S(LN)2 (**a**, **c**, **e**) or 100% 2-3S(LN)2 (**b**, **d**, **f**). Statistical significance (*P*) was calculated by two-tailed unpaired *t* test on biological replicates (**a**, **b** $n = 6$; **c-f** $n = 3$). (*$P = 0.0493$; **$P = 0.0085$; ***$P = 0.0005$; ****$P < 0.0001$). Data are presented as mean values $+/-$S.D. Panel **g** was Created with BioRender.com. Source data are provided as a Source Data file.

entry. We genetically modified the sialylation pattern of a cell line that was used for virus entry as well as production of a sialoglycoprotein receptor used for virus binding. The Sia repertoire of HEK$^{\Delta Sia}$ cells, depleted of endogenous Sia expression by deletion of all ST6Gal and ST3Gal isoenzymes[32], was specifically tuned by transient expression of sialyltransferases ST6Gal1 and/or ST3Gal4. These two transferases specifically sialylate the terminal LacNAc extensions that are especially abundant on N-linked glycoproteins. To enable virus binding analysis by BLI on a sialoglycoprotein receptor with a similar sialylation pattern as these transfected HEK$^{\Delta Sia}$ cells, the luminal domain of lysosomal-associated membrane glycoprotein I (LAMP I, carrying 18 N-linked glycans) was expressed in these cells. It is important to note that virus binding specificity for LAMP I carrying either 2-3Sias or 2-6Sias differs substantially from the pattern obtained on synthetic glycan receptors (Fig. 3b, c). HU02 (H5N1) is absolute 2-3Sia specific and BK79 and WU95 (H3N2) are absolute 2-6Sia specific on protein and synthetic glycan receptors.

However FU02 (H3N2), which is 2-6Sia specific on synthetic glycans, does not bind to LAMP I at all. VI75 is dual-specific on LAMP I but mostly binds 2-6Sia on synthetic glycans. PR8 is relatively 2-3Sia-specific on both receptor types but displays moderate 2-6Sia binding on both.

HEK cells and LAMP I containing a range of Sia densities (2-3Sia, 2-6Sia or mixtures of both at different ratios) were obtained by transfection of HEK$^{\Delta Sia}$ cells with varying amounts of plasmids encoding the sialyltransferases ST6Gal1 and/or ST3Gal4. Sia content of the differentially sialylated LAMP I batches was quantified by plotting lectin binding (SNA, 2-6Sia specific; MAL I, 2-3Sia specific) against the amount of sialyltransferase that was used. Data from 4 to 6 biological replicates were assembled in a single plot showing the high reproducibility of amount of sialylation at a specific expression condition (Supplementary Fig. 4). Clearly, ST3Gal4 and ST6Gal1 compete for terminal galactose substrates, as shown by reduction of 2-6Sia levels (detected by SNA binding) when a ST6Gal1

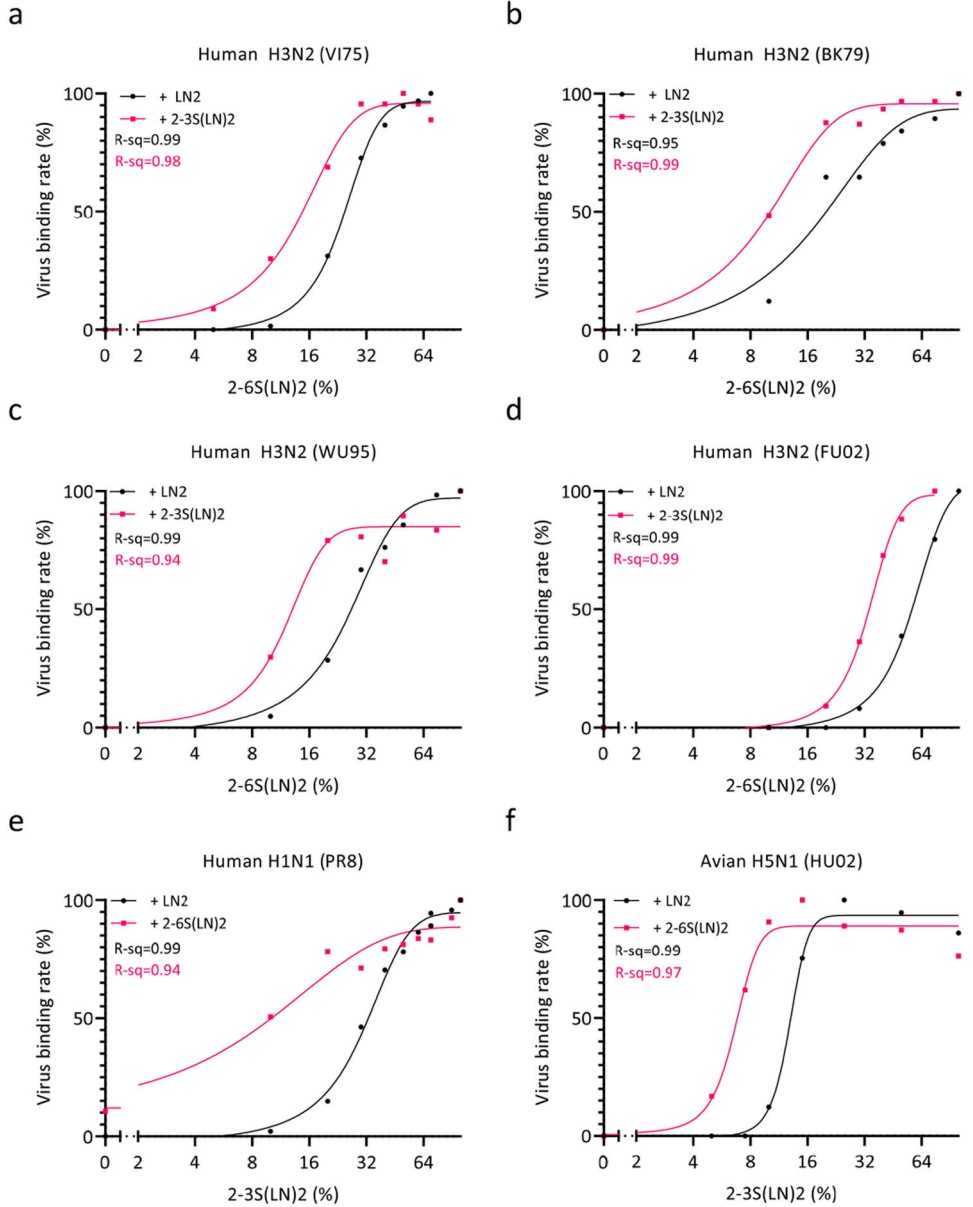

**Fig. 2 Low-affinity receptors lower the threshold density for IAV binding by hetero-multivalent interactions to a heterogenous receptor surface.**
Relative virus binding rates were plotted as a function of relative receptor density for human H3N2 strains VI75 (**a**), BK79 (**b**), WU95 (**c**) and FU02 (**d**), human H1N1 strain PR8 (**e**) and avian H5N1 strain HU02 (**f**). The relative density of the high-affinity receptor is plotted on the x-axis. The black curves display binding to the high-affinity receptor complemented to 100% density with the asialo glycan LN2 (Galβ1-4GlcNAcβ1-3Galβ1-4GlcNAc) whereas the magenta curves display binding to the high-affinity receptor complemented to 100% density with the low-affinity glycans. Relative binding rates are calculated for each point from individual binding curves for each density (see Supplementary Fig. 1 for an example). The absolute binding rates of the different virus strains are plotted in Fig. 3c showing that they differ less than 2-fold. Typical examples from 3 biological replicates are shown for each virus. R-sq values for fitted curves are indicated. Source data are provided as a Source Data file.

concentration range is co-expressed with increasing amounts of ST3Gal4 (Supplementary Fig. 4c). In conclusion, Sia density of complementary 2-6Sia/2-3Sia gradients expressed on LAMP I can be quantified by lectin staining. Therefore, plotting lectin staining against virus binding will allow to identify hetero-multivalent binding effects for IAV binding to natural glycoprotein receptors.

**Hetero-multivalent IAV binding to the sialoglycan protein receptor LAMP I**. The binding rate of 2-6Sia-specific human IAV strains BK79 and WU95 to a series of LAMP I proteins displaying increasing amounts of 2-6-linked Sia was determined by BLI (Fig. 3d, e). 2-6Sia was complemented by 2-3Sia (magenta lines)

or by asialoglycan (black lines). The presence of 2-3Sias on LAMP I enhanced the binding rate of BK79 and WU95 as shown by the leftward shift of the magenta lines relative to the black lines. Thus, in the presence of 2-3Sia moieties, a lower density of 2-6Sia on LAMP I is required to be able to support binding. Such a positive hetero-multivalent binding effect is also seen for 2-3Sia specific virus strains HU02 (H5N1) and PR8 (H1N1) where in this case 2-3Sias are complemented by 2-6Sia. Note that for all four strains (shown in Fig. 3d–g) no binding was observed when LAMP I was decorated with the low-affinity Sia only. We conclude that positive hetero-multivalent binding effects assist binding of H3N2, H1N1 and H5N1 to sialoglycoprotein receptors.

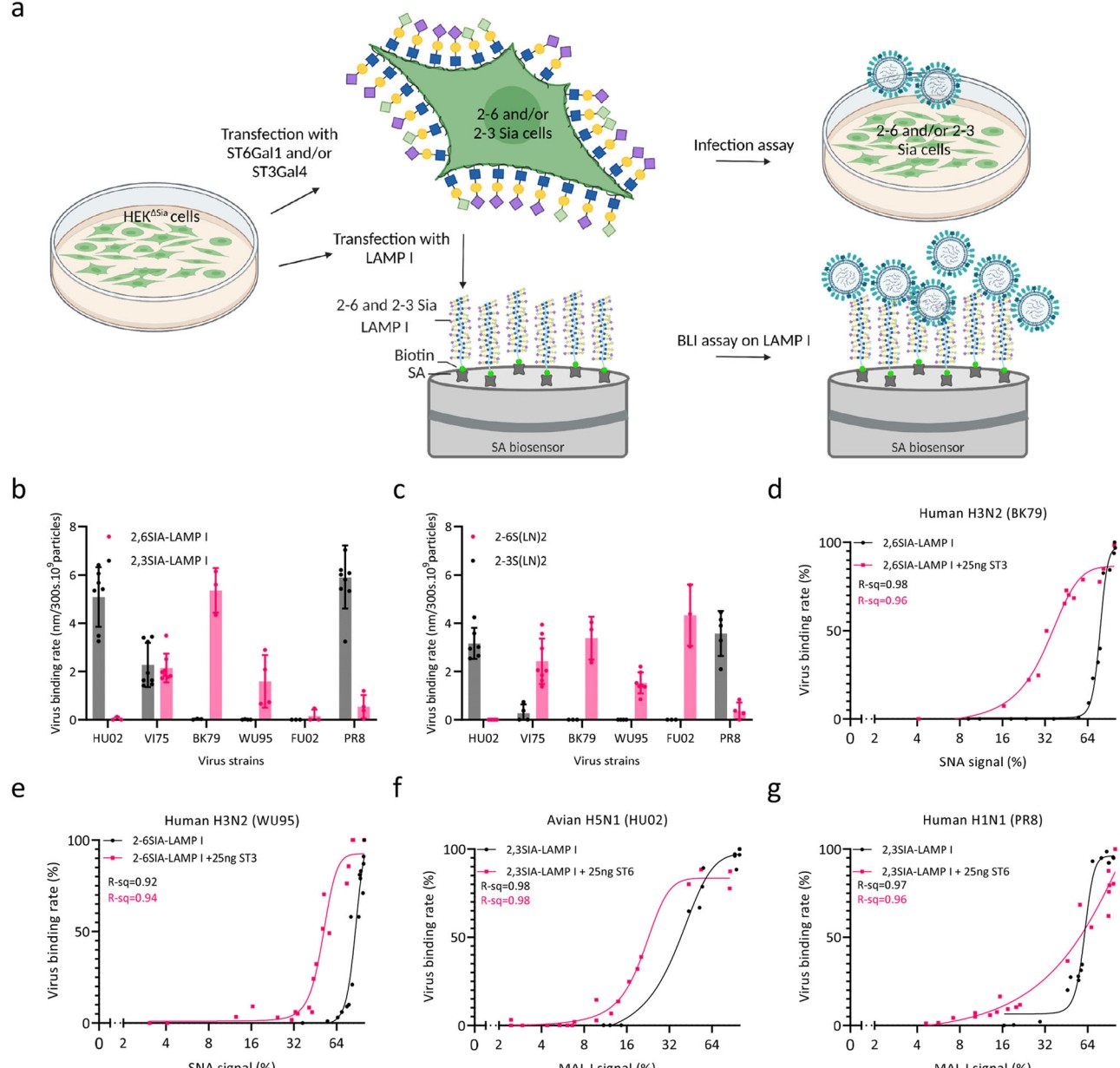

**Fig. 3 Hetero-multivalent binding to sialoglycoprotein LAMP I decorated with complementary gradients of 2-3Sia and 2-6Sia. a** Graphical summary of methods that were employed to establish density gradients on HEK$^{\Delta Sia}$ cells and LAMP I. The Sia repertoire exposed on HEK$^{\Delta Sia}$ cells was specifically tuned by transfection with different concentrations of ST6Gal1 and/or ST3Gal4. Co-transfected LAMP I will be secreted in a C-terminally biotinylated form that can be attached to streptavidin BLI sensors for comparing entry into HEK$^{\Delta Sia}$ cells with binding to LAMP I with a similar sialylation pattern in a BLI assay. **b**, **c** Comparison of virus binding rates to avian-type and human-type receptors. Absolute binding rates (nm/300 s), normalized to a particle number of $10^9$ particles of each virus strain, were determined. **b** Binding to sensors fully loaded with LAMP I produced in HEK$^{\Delta Sia}$ cells co-transfected with 25 ng ST6Gal1 or 25 ng ST3Gal4 yielding high levels of LacNAc termini carrying 2-6Sia (magenta bars) or 2-3Sia (black bars). **c** Binding to sensors fully loaded with 2-6S(LN)2 (magenta bars) or 2-3S(LN)2 (black bars). Relative binding rates were plotted as a function of relative receptor density for human H3N2 strains BK79 (**d**) and WU95 (**e**), avian H5N1 strain HU02 (**f**) and human H1N1 strain PR8 (**g**). Black lines show binding to LAMP I proteins obtained by co-expression with a concentration range of sialyltransferases ST6Gal1 (**d**, **e**) or ST3Gal4 (**f**, **g**). Magenta lines show binding to LAMP I proteins obtained by co-expression of a concentration range of ST6Gal1 in combination with 25 ng ST3Gal4 (**d**, **e**) or a concentration range of ST3Gal4 in combination with 25 ng ST6Gal1 (**f**, **g**). 2-6Sia density or 2-3Sia density on LAMP I was quantified by SNA or MAL I binding respectively and plotted on the x-axis as relative SNA (**d**, **e**) or MAL I (**f**, **g**) signal reflecting the relative receptor density. Data from biological replicates were fitted into single curves of which R-sq values are indicated. (**b** n = 7, 3, 8, 8, 3, 3, 4, 4, 3, 3, 9 and 4 for each bar from left to right; **c** n = 6, 6, 4, 8, 3, 3, 4, 7, 3, 3, 5 and 4 for each bar from left to right). Data are presented as mean values +/−S.D. Panel **a** was Created with BioRender.com. Source data are provided as a Source Data file.

## The effect of sialic acid receptor density on virus infection efficiency.

Binding avidities derived from in vitro binding assays often do not correlate quantitatively with virus entry efficiency into cell line cultures. Likely, virus binding dynamics to the highly complex glycocalyx of cells involves hetero-multivalent interactions with a diverse range of Sia receptors of unknown density and distribution on the cell surface including O-linked glycans and glycolipids. Specific tuning of the Sia receptor repertoire on the

surface of HEK$^{\Delta Sia}$ cells by co-transfection of ST6Gal1 and ST3Gal4 offers unique opportunities to assess the effect of hetero-multivalent binding on cell entry using a luciferase reporter assay[34]. We first determined the dependency of virus infection efficiency on the density and specificity of Sia receptors on the surface of HEK$^{\Delta Sia}$ cells that were co-transfected with a concentration range of sialyltransferases ST6Gal1 or ST3Gal4. Clearly, infection was highly specific for the expressed Sia linkage-type for human

H3N2 strains BK79 and WU95 (2-6Sia) and for avian strain HU02 (2-3Sia, Fig. 4a–c). PR8 was slightly less specific as transfection with high amounts of ST6Gal1 supported efficient entry (Fig. 4d). Entry by Sia-independent enterovirus CVB3 was identical in all (trans-fected) cell types showing, as a control for cell integrity, that the differential display of sialyltransferases specifically affects IAV entry (Supplementary Fig. 5). Entry of HU02 into HEK$^{\Delta Sia}$ cells and HEK$^{\Delta Sia}$ cells transfected with ST6Gal1 showed the same basal

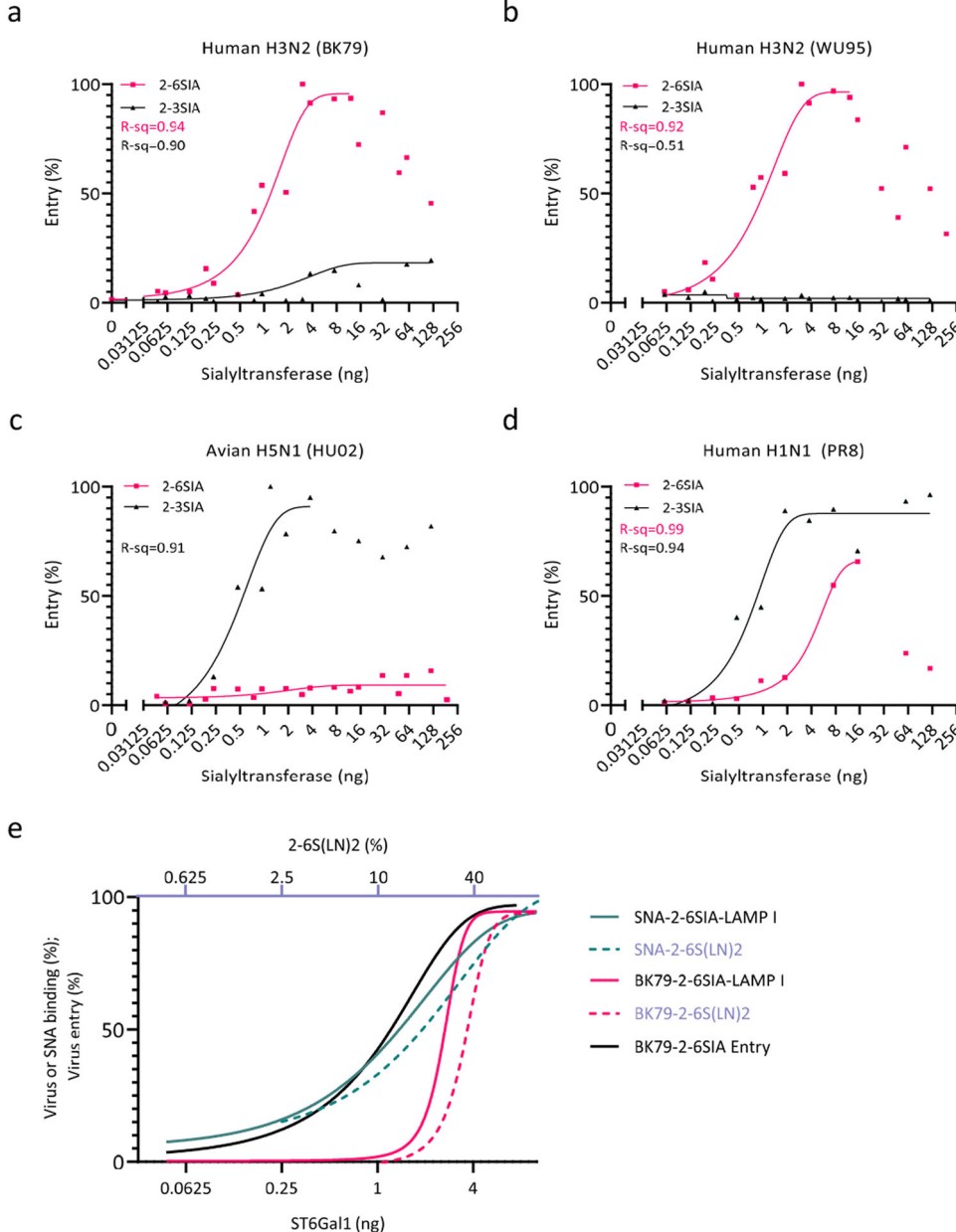

**Fig. 4 Low-affinity receptors contribute to virus entry efficiency. a–d** Virus entry efficiency into HEK$^{\Delta Sia}$ cells transfected with a concentration range of sialyltransferases ST6Gal1 (magenta lines) or ST3Gal4 (black lines) was determined using a luciferase reporter and a single-round infection assay. Relative entry efficiency into cells (normalized to the highest value) was plotted as a function of transfected sialyltransferase quantities. Data are from four biological replicates and each point represents the mean of 3 measurements. Entry often reached a maximal efficiency followed by a decline upon further increase of transfected sialyltransferase concentration. Data points beyond maximal entry efficiency were not fitted to the curves. **e** Comparison of virus binding and entry kinetics. Relative entry efficiency (black line) and binding rate to LAMP I (solid magenta line) or 2-6 S(LN)2 (dotted magenta line) of human H3N2 strain BK79 was plotted as a function of receptor density. The lower x-axis shows the amount of ST6Gal1 that was used for transfection of HEK$^{\Delta Sia}$ cells into which entry was performed or in which LAMP I glycoprotein receptor was expressed. Top x-axis shows the density of 2-6S(LN)2 receptor as percentage of the maximal density of 0.6 pM/cm$^2$ (corresponding to ~0.0036 Sia moieties per nm$^2$ [23]). BLI binding data for BK79 come from experiments shown in Fig. 2b (2-6S(LN)2) and Fig. 3d (LAMP I). In addition relative SNA binding (performed to the same sensors as used for virus binding) was plotted in parallel (teal lines). Entry data for BK79 were derived from the experiment shown in Fig. 4a. Source data are provided as a Source Data file.

level of entry that potentially represents some SIA-independent entry (Supplementary Fig. 5). IAV infection efficiency was dependent on the amount of sialyltransferase that was used for transfection consistent with the FACS data of HEK$^{\Delta Sia}$ cells transfected with sialyltransferases, which showed that the density of cell surface Sia correlates with the amount of transfected sialyltransferase (Supplementary Fig. 6). Remarkably, a maximal infection efficiency was reached for 2-6Sia-dependent entry, followed by a decline upon further increase of transfected ST6Gal1 dose (Fig. 4a, b). Likely, this relates to other factors known to affect virus binding efficiency, like for instance the number of LacNAc repeats of a sialylated glycan antennae[16]. Capping of glycan chains by sialic acid will block the addition of additional sugar moieties and therefore glycan chain length is dependent on the level of sialyltransferase activity.

To directly compare how receptor density affects virus binding and entry we re-plotted the binding and entry results for BK79 obtained above in parallel on the same scale (Fig. 4e). BK79 entry efficiency (Fig. 4e, black line) and BK79 binding rate to LAMP I (Fig.4e, solid magenta line) are plotted against ST6Gal1 on the lower x-axis and compared to BK79 binding to 2-6S(LN)2 (dotted magenta line) plotted against receptor density on an identical logarithmic scale (upper x-axis). The steep slope of the BK79 binding curves is reflecting the super-selective binding mode that is inherent to the binding of multivalent particles to a receptor surface (see discussion) and contrasts with the shallower binding curves for bivalent SNA lectin binding (teal lines) observed on LAMP I and synthetic glycans. Hetero-multivalent interactions with low-affinity receptors further decrease the threshold-density of the high-affinity receptor required for binding, still supporting a super-selective binding mode (see the similar slope of the hetero-multivalent binding curves in Figs. 2 and 3 in comparison to lines obtained on homogeneous receptors). Entry into cells (Fig. 4e, black line) is detected at lower ST6Gal1 doses than binding to LAMP I. Super-selective virus binding translates into efficient virus entry above the receptor threshold as the slope of the BK79 entry curve is only slightly less steep than for BK79 binding. Although the highly complex cell surface may harbor patches of high glycan-density that are lacking on LAMP I, we conclude that very low densities of the specific receptor efficiently support virus entry.

**The effect of hetero-multivalent IAV binding on IAV infection efficiency.** To determine effects of hetero-multivalency on virus entry, suboptimal concentrations of ST6Gal1 and ST3Gal4 (i.e. supporting only very limited entry) were selected from the graphs in Fig. 4a–d and transfected in combination into cell line HEK$^{\Delta Sia}$. After 24 h cells were infected with IAV strains BK79, WU95, HU02 and PR8 (Fig. 5). An ST3Gal4 concentration-dependent increase of infection by 2-6Sia specific strains BK79 and WU95 was observed in combination with the highest concentration of ST6Gal1 (0.8 ng). Similarly, an ST6Gal1 concentration-dependent increase of infection by 2-3Sia specific strains HU02 and PR8 was observed in combination with the highest concentration of ST3Gal4 (0.2 ng). Thus, at a relatively low concentration of the high-affinity receptor, entry is enhanced by the presence of a Sia receptor that by itself only supports very low levels of entry. Importantly, all entry experiments were performed in the absence of the neuraminidase (NA) inhibitor Oseltamivir Carboxylate (OC). Therefore, we conclude that hetero-multivalent binding effects enhance entry efficiency of IAV and can be observed under natural conditions where NA is active.

## Discussion

Over 400 (sialo) glycans have been identified on respiratory epithelium[8,35–37]. A range of assays previously identified many of them as non-binders for particular IAV strains with the α2-3/α2-6 dichotomy in avian versus human IAV receptor preference as the most striking result. Here we show that such presumed non-binders can play an important role in IAV binding and subsequent cell entry via a hetero-multivalent binding mode that has hitherto only been described for lectins and glycosyltransferases[28–31]. We first identified hetero-multivalent IAV binding using heterogeneous sets of synthetic glycans loaded on BLI sensors. Subsequently, we designed a quantitative method for genetic tuning of the glycan composition on HEK cells and a secreted glycoprotein. Using these, we showed that hetero-multivalent binding enhances IAV entry efficiency. These findings challenge the longstanding point of view that avian-type 2-3Sia receptors hardly contribute to infection by human IAVs that require 2-6Sias for binding, and vice versa. This emphasizes the need to consider virus binding in the context of the receptor diversity and distribution on a relevant target tissue.

We propose a model for hetero-multivalent binding as shown in Fig. 6a. Because of the low affinity of HA-Sia interactions ($K_D$ is ~1 to ~20 mM[18–20]), even the initial monovalent interaction between a viral HA and its preferred receptor (left panel, receptor-associated HAs are indicated in magenta) is very short-lived (half-life less than a few seconds). As a result, the balance is towards dissociation (black arrows). Still, some particles will form a second interaction, resulting in bivalent binding. The $K_D$ ($= k_{off}/k_{on}$) of the initial interaction, as well as receptor density, are major factors that determine the frequency of bi-valent complex formation which, by its much higher avidity, promotes engagement of additional receptors. The equilibrium of an initial interaction with a non-preferred receptor of relatively low affinity (right panel) will however be further to the unbound state, thereby almost completely preventing a transition to bivalent binding. In contrast, at a surface of mixed low- and high-affinity receptors (middle panel), low-affinity receptors can function in the formation of the second and subsequent interactions leading to increase of avidity by hetero-multivalent binding, even though they are not able to initiate high-avidity virus binding by themselves (further details can be found in the legend to Fig. 6). In summary, a high-affinity primary sialoglycan attachment receptor is required to initiate stable multivalent binding to a heterogeneous receptor-landscape that can be largely composed of a range of low-affinity sialoglycan receptors. Clearly, the canonical 2-3Sia/2-6Sia binding dichotomy of human versus avian IAVs is far from absolute.

Multiple reports have described the presence of a secondary sialic acid binding site (2SBS) on NA[38,39] as well as a secondary sialic acid binding site, called the vestigial esterase subdomain (VES), on HA[18]. The 2SBS is conserved in avian, but not in human IAVs[40–42], and displays 2-3Sia specificity although a 2SBS of much lower affinity was suggested to be present in some pandemic H1N1 strains[43]. Binding affinity for the putative VES was estimated to be ~200 mM[44]. Such potential binding sites of low affinity may expand the options of forming different hetero-multivalent interaction networks as illustrated in Fig. 6b and is further described in the legend to Fig. 6b. These diverse and highly dynamic interaction networks may in turn affect the efficiency of NA to associate with, and cleave, Sias, thereby influencing virus motility and dissociation (Fig. 6c).

The steep sigmoidal IAV binding curves (Figs. 2 and 3) correspond with a super-selective binding mode of multivalent ligands to a receptor surface[45,46]. Super-selective binding to high affinity receptors was observed for homogeneous and heterogeneous receptor surfaces, even when the monovalent $K_D$ for high-affinity receptors (e.g., 2-3Sias) is only slightly lower than low-affinity receptors (e.g., 2-6Sias) that do not bind by themselves. Importantly, low-affinity receptors can lower the density

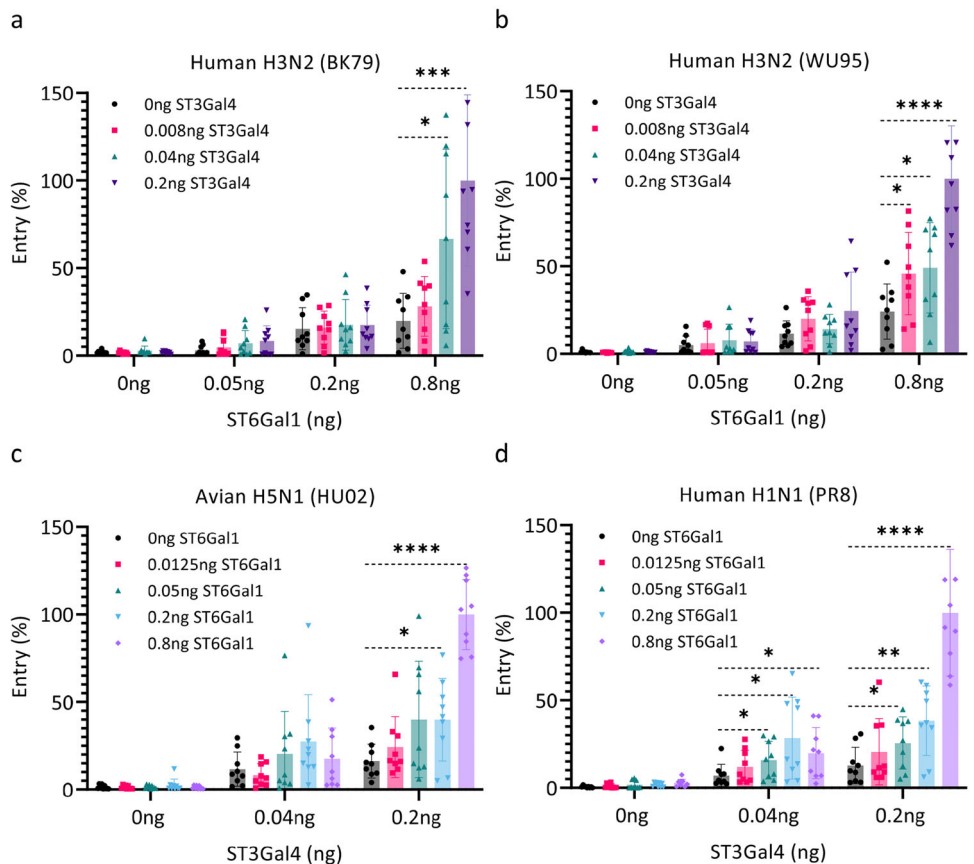

**Fig. 5 Coinfection of low-affinity receptors enhances IAV infection. a–d** HEK$^{\Delta Sia}$ cells were transfected with the indicated combinations of ST3Gal4 and ST6gal1. The transfected amount of the high-affinity receptor is indicated on the x-axis and of the low-affinity receptor by the color of the bars. Note that co-expressed ST3Gal4 and ST6Gal1 are in competition (Supplementary Fig 3C). Bars represent the average of 3 biological replicates that are all performed in triplicates. Statistical significance (*P*) was calculated by two-tailed unpaired *t* test on biological replicates (*n* = 9). (**a** ***$^{***}P$ = 0.0003, $^{*}P$ = 0.0185; **b** $^{****}P$ < 0.0001, $^{*}P$ = 0.0353, $^{*}P$ = 0.0246; **c** $^{****}P$ < 0.0001, $^{*}P$ = 0.0136; **d** 0.04 ng group, $^{*}P$ = 0.0294, $^{*}P$ = 0.0175, $^{*}P$ = 0.0149; 0.2 ng group, $^{****}P$ < 0.0001, $^{**}P$ = 0.0036, $^{*}P$ = 0.0193 from top to bottom). Data are presented as mean values +/−S.D. Source data are provided as a Source Data file.

threshold required for binding to high-affinity receptors (Figs. 2 and 3). Strikingly, virus entry also rapidly increases above a threshold receptor density (Fig. 4e) suggesting that, in contrast to what has been frequently reported[8,11–14], there is a strong positive correlation between binding avidity and entry. The strict correlation between binding and entry is further corroborated by the enhancement of entry efficiency under conditions that support hetero-multivalent binding (Fig. 5). Importantly, different respiratory tissues are often highly enriched for either 2-3 or 2-6 sialosides but neither of them is ever completely absent giving ample opportunity for a hetero-multivalent binding mode in vivo. Specific tuning of the glycan receptor repertoire of HEK$^{\Delta Sia}$ cells has been of critical importance for identifying the positive effect of hetero-multivalent binding on IAV entry. The limited receptor diversity resulting from sialylation of Galβ1,4GlcNAc termini by ST6Gal1 and/or ST3Gal4 on HEK$^{\Delta Sia}$ cells still contrasts, however, with the complex glycan repertoire exposed on epithelial cells. This receptor complexity, which is hardly assessed by current binding assays, can be further dissected and correlated to entry by the receptor tuning approach taken here.

Hetero-multivalent interactions have implications for virus entry, tropism and evolution. Receptor-bound IAV particles are highly motile[22,47] and explore the cell surface for minutes before entry[48,49]. This NA activity-driven directional motility and entry efficiency depends on balanced NA and HA activity[23,47,50]. NA activity may be affected by hetero-multivalent interactions as, like HA, NA displays receptor specificity. Avian IAVs, engaging 2-3

Sias for primary attachment, preferentially cleave 2-3Sias over 2-6Sias (Fig. 6c, $k_{cat1} > k_{cat2}$). Upon primary attachment to a heterogeneous receptor surface via high-affinity receptors, migration to poorly cleavable 2-6Sia-rich clusters could reduce motility which has been observed to precede cell entry[48,49]. IAV particle binding-induced clustering of sialylated receptors at such spots[51] is signaling induction of their uptake by endocytosis[34,48,51–56]. High-avidity binding via low-affinity interactions has been shown to efficiently induce clustering and signaling in other systems[57]. Thus, signaling receptors decorated with Sias with either low or high affinity could be clustered by any particular IAV strain translocating over the cell surface upon binding initiated by hetero-multivalent interactions at a distant spot. In summary, entry efficiency can directly be enhanced through hetero-multivalent interactions, increasing binding rate and possibly exerting effects on virus motility, receptor clustering and signaling.

Hetero-multivalent binding likely also affects cell-type specificity as well as within-host evolution. We here showed that seasonal IAVs displaying a 2-6Sia binding preference, could infect cell types displaying a very low 2-6Sia density by hetero-multivalent binding (and likewise for avian IAVs and 2-3Sias). Such cells are present in the 2-3Sia-rich lower parts of the human respiratory tract, a place where IAV infection more easily causes enhanced pathogenicity like pneumonia. Recent H3N2 strains have evolved to bind 2-6Sias attached to extended chains of LacNAc repeats on multi-antennary glycans and display decreased avidity for the more abundant shorter glycans[12,16].

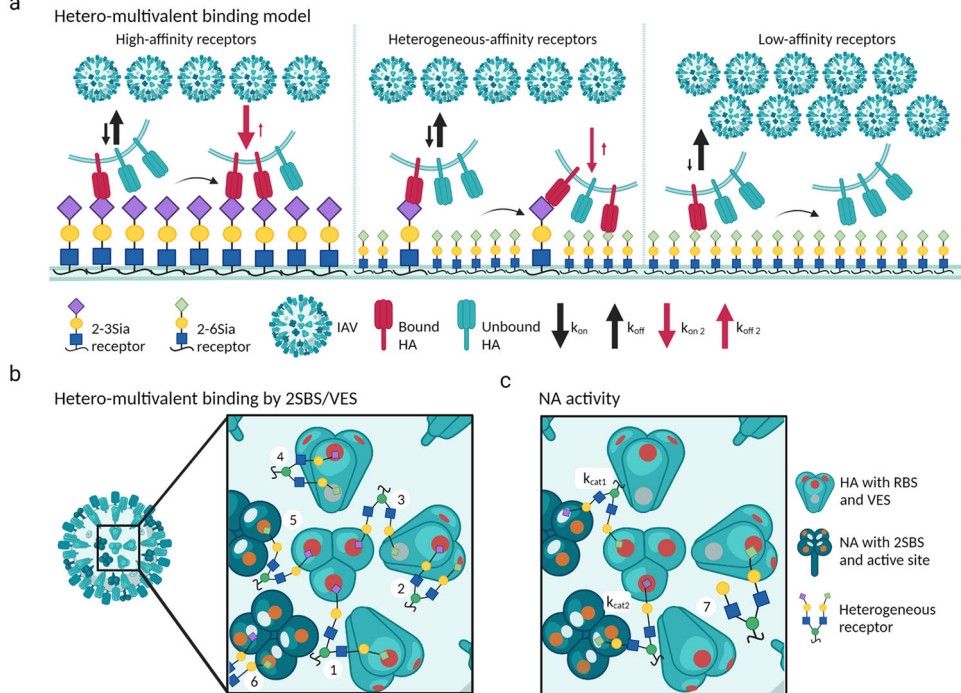

**Fig. 6 Model for hetero-multivalent IAV binding. a** Avian IAV interactions with high-affinity (2-3Sia), low-affinity (2-6Sia) or mixed receptor surfaces. Arrow size indicates how $k_{on}$ and $k_{off}$ relatively compare. The initial, intermolecular and virus concentration dependent, monovalent interaction ($K_D = k_{off}/k_{on} \sim$ 1–20 mM) is characterized by a low binding rate constant $k_{on}$ ($M^{-1} s^{-1}$) and high dissociation rate constant $k_{off}$ ($s^{-1}$, half-life time of 0.5 to a few seconds)[17-21]. Additional interactions will lead to longer lasting virus-to-surface binding. The second, concentration-independent, interaction depends on the $k_{off}$ of the first interaction and a second $k_{on2}$ that depends on the nature of, and accessibility to, a second glycan. A $k_{off2}$ for virus dissociation is much lower than $k_{off1}$ as it describes a bivalent interaction. This initiates a cascade of sequentially faster engagement of subsequent receptors. Thus, IAV binding rate primarily depends on an initial high-affinity interaction with a 2-3Sia receptor that determines the chance of forming additional interactions with high-affinity 2-3Sia receptors (left panel) or by hetero-multivalent interactions with low-affinity 2-6Sia receptors (middle panel). However, at a homogeneous 2-6Sia receptor surface (right panel), the $K_D$ for the initial interaction in combination with a 2nd $k_{on}$ for a 2-6Sia receptor will not support multivalent binding (note that reported $K_D$ values for low and high affinity receptors for IAV strains differ only ~1.5 to 3-fold[19]). **b** Hetero-multivalent binding via the HA receptor binding site (RBS, magenta), a vestigial esterase site (VES, grey) and a secondary Sia binding site (2SBS, orange) on NA. As an example potential interactions for a bi-antennary receptor carrying a 2-3Sia (purple) and 2-6Sia (light green) are shown: inter-trimeric (1) and intra-trimer binding (2) of the RBS; inter-trimeric (3) and intra-trimer (4) binding via the RBS and VES; HA-NA binding via RBS and 2SBS (5). **c** NA activity effects on heterogeneous receptor surfaces. NA catalytic site (white) activity depends on substrate structure, being higher on 2-3Sia ($k_{cat1}$) than on 2-6Sia ($k_{cat2}$). Catalytic activity can be affected by arrangement of receptors via the different binding sites, for instance by binding to 2SBS like in (6). NA activity drives virus motility by reducing receptor density (7). Created with BioRender.com.

Also, a 2-3Sia-binding pH1N1 virus rapidly evolved to airborne transmission in ferrets by acquiring binding to long-chain 2-6Sias, most likely being selected on the soft palate that presents these receptors as well as 2-3Sia receptors[58]. Hetero-multivalent binding may enable the evolution of such viruses by retaining some affinity for the abundant "old" receptor. In that case, only a low-density of the new high-affinity receptor will be required.

Little is known about the evolutionary track leading to adaptation to novel host receptor repertoires. Early isolates of four IAV pandemics already acquired a large specificity-switch to a human 2-6Sia receptor. Although this switch requires only two mutations in HA, the infrequent occurrence of IAV pandemics is attributed to the requirement for simultaneous mutations in multiple segments, collectively enabling human-to-human transmission. Avian strains H5N1 and H7N9, causing severe infections of the 2-3Sia-rich human lower respiratory tract, did not acquire human-to-human transmission in 25 years. Likely, acquisition of increased 2-6Sia-binding has no selective advantage in the lower respiratory tract. However, 2-6Sia-rich respiratory surfaces with a minimally required 2-3Sia density can support hetero-multivalent binding and entry of avian IAVs resulting in selective advantages for mutations that increase 2-6Sia binding. Such mutations can function as stepping stones towards the

genesis of a human-transmissible virus. Intermediate host species may provide such an environment. Indeed, whereas human nasal epithelial cells - a favorable location for human transmissible IAVs[59] - are almost exclusively decorated with 2-6Sia, a gradient of 2-3Sia concentrations along the 2-6Sia-rich tracheal surface of swine was demonstrated to occur[60]. In line herewith, a gradual shift to the use of 2-6Sia receptors in swine has been demonstrated very recently for an avian-derived swine IAV lineage[61]. Hetero-multivalent interactions with such areas on the respiratory epithelium, as well as with decoy receptors present in the overlaying mucus, could be a key factor in the evolutionary pathways leading to host adaptation.

## Methods

**Virus strains.** The human H3N2 recombinant influenza strains used in this paper were generated and described previously[62]: A/Bilthoven/1761/76 H3N2 (GenBank accession no. ADT78971) (VI75); A/Netherlands/233/82 193 N (GenBank accession no. AFH00725) (BK79); A/Netherlands/178/95 145 K (GenBank accession no. AIU46036) (WU95); A/Netherlands/213/03 (GenBank accession no. AIU46062) (FU02). H1N1 strain A/Puerto Rico/8/34/Mount Sinai (H1N1, PR8) was also used. Recombinant H5N1 (referred to as HU02 WT in this paper) contains the NA gene (GenBank accession no. BAM85820.1) and the HA gene (GenBank accession no. ACA47835.1) from A/duck/Hunan/795/2002(H5N1) in the background of A/Puerto Rico/8/34/Mount Sinai (H1N1, PR8). Cell culture adaptation of this strain leads to the stable introduction of mutation A156T in H5 after a few

passages. This mutation introduces a glycosylation site at position 154 as is present in many contemporary H5N1 field isolates (e.g., ABC66574.1) and is referred to as HU02. Recombinant H5N8 influenza strain carries the HA gene of strain A/Chicken/Netherlands/14015526/2014 (GISAID accession no. EPI_ISL_167905) in the background of 7 PR8 segments. Virus strains were propagated by inoculating seed stocks into Madin-Darby canine kidney (MDCK)-II as described[34] and stored at −80 °C. Enterovirus strain RlucCVB3 was obtained by transfection of infectious clones pRluc-53CB3/T7[63] and stored in lab, MOI = 0.1 was used in infection and luciferase reporter assay.

**Nanoparticle tracking analysis (NTA)**. Virus particle numbers were determined by Nanoparticle Tracking Analysis (NTA) using a NanoSight NS3000 (Malvern Panalytical) as described for quantifying IAV particle numbers[42]. Briefly, 100 ul of virus samples were loaded into the NTA chamber five times independently. If necessary, virus samples were diluted in PBS before measurement and analyzed again. Particle numbers were determined as the average of five measurements. Multiple independent batches of each virus strain were measured for biological replicates of experiments. Particle numbers of all virus batches used in this paper were within $1.0 \times 10^9$ and $5.0 \times 10^9$ virus particles/ml. Virus particle numbers were used to normalize virus binding rate in binding assays and normalize infection levels among virus strains in infection assays.

**Expression and purification of biotinylated glycoprotein receptors**. All recombinant glycoproteins were expressed in a HEK293-derived knock-out cell line HEK$^{\Delta Sia}$ [64], that lacks all Siaα2-3Gal and Siaα2-6Gal glycotopes on cell surfaces as eight sialyltransferases (ΔST3GAL1/2/3/4/5/6 and ST6GAL1/2) were knocked out. To specifically tune sialylation patterns on a secreted recombinant, C-terminally biotinylated, glycoprotein, specific sialyltransferases encoding expression vectors (pCDNA-ST3Gal4 and/or pCDNA-ST6Gal1) (Genscript, Piscataway, NJ, USA, NM_1254757.2; NM_173216.2) and an expression vector (pCD5) encoding a BirA biotin ligase[65] were co-transfected with an expression vector coding for the extracellular domain of lysosomal membrane protein I (LAMP I) fused to a C-terminal biotinylation site. A codon optimized glycoprotein LAMP I ectodomain-encoding cDNA (Genscript, Piscataway, NJ, USA) was cloned in pCAGGS in frame with a C-terminal 2xStrep-tag followed by a biotin-acceptor peptide (BAP, GGLNDIFEAQKIEWH) for site specific biotinylation by BirA, allowing binding to streptavidin (SA) biosensors, and a 6xHis tag for purification using Ni-NTA beads. The HEK$^{\Delta Sia}$ cells were seeded in cell culture flasks or 6-well plates in culture medium DMEM (Gibco), supplemented with 10% fetal bovine serum (FBS) (Biowest), 10 U/ml of PEN-STREP (Lonza) and 2 mM Glutamax (Gibco) at 37 °C and 5% CO$_2$ one day before transfection. Plasmids were transfected at different ratio's (as indicated in the text) into HEK$^{\Delta Sia}$ using polyethyleneimine I (PEI, Polysciences) as described[23]. Briefly, the transfection mixture containing DNA and PEI in 1:10 ratio were mixed rigorously and incubated 30 min at room temperature. The transfection mixture was added carefully (drop-by-drop) to cell culture flasks containing half-volume of cell culture medium without PEN-STREP. The biotinylated LAMP I was purified from cell culture supernatant 84 h after transfection according to standard protocols of Ni-NTA purification system (ThermoFisher Scientific K95001).

**Real time virus binding and lectin binding assays by Biolayer interferometry (BLI)**. Real time virus binding was studied by BLI analysis using an Octet RED348 (Fortebio). All experiments were carried out at 30 °C in Dulbecco's PBS with Calcium and Magnesium (PBS + / + ) (Lonza) as standard assay buffer. Streptavidin (SA) biosensors were loaded with Sia receptors using biotinylated synthetic glycans that contain either α2-3 Sia, α2-6 Sia or no Sia with two LacNAc (N-acetyllactosamine [Galβ1-4GlcNAc]) repeats (referred to as 2-3S(LN)2, 2-6S(LN)2, and LN2 in this paper), or were loaded with specifically glycosylated recombinant glycoprotein LAMP I batches as indicated in the text. Receptors were loaded to densities indicated in the text or Figures. Different loading densities were obtained by loading a single receptor for different loading times and/or at different receptor concentrations. Receptor densities (loading levels) were determined from the BLI signal and plotted relatively to the BLI signal obtained at full loading. Alternatively, mixed receptor surfaces were obtained by mixing two or three receptors at the required ratios and loading the mixture to full occupation of the BLI SA-biosensor surfaces. Individual receptors were first exactly calibrated to each other on basis of their binding rate. Real time virus association was examined for 900 s by moving receptor-loaded biosensors to wells containing 100 ul virus sample in the presence of 10 μM Oseltamivir Carboxylate (OC; Roche) to block all NA activity[23], 1 mM Protease inhibitor (Roche; 11836153001) and Trypsin-like proteases inhibitor (Sigma Aldrich; T6552). Virus initial binding rates were calculated from virus binding levels after 5 min (during which the binding curve is linear apart from the first few seconds). Initial binding rates were either plotted absolute (nm/10$^9$ virus particles) or relative to the maximum binding rate at full receptor density as indicated in the Figures. LAMP I sialylation levels were analyzed by lectin binding assays by incubation of receptor-loaded biosensors in lectin-containing wells (4 ug lectins/well) and recorded by BLI for 900 s to reach equilibrium lectin binding levels. *Maackia amurensis* lectin I (MAL I, Vector Labs) binds to α2-3Sia-Galβ1-4GlcNAc, *Sambucus nigra* lectin (SNA, Vector Labs) binds preferentially to α2-6Sia-Galβ1-4GlcNAc and *Erythrina Cristagalli* Lectin (ECA, Vector Labs) binds to terminal LacNAc (Galβ1-4GlcNAc).

**Infection and luciferase reporter assays**. For determining the virus infection efficiency in correlation to the binding properties determined by BLI, virus infection assays were applied in specifically sialylated HEK$^{\Delta Sia}$ cells as described in the text. A luciferase reporter system[66] employing the Gaussia luciferase encoding vector pHH-Gluc was used as described[34] for performing a single-round entry assay. HEK$^{\Delta Sia}$ cells were seeded in a 96-well plate (20.000 cells/ well) to reach 90% confluency the following day at which cells were cotransfected with 50 ng pHH-Gluc and a range of sialyltransferases (ST3Gal4 and/or ST6Gal1) using Lipofectamine 2000 (InVitrogen) according to the manufacturer's protocol. The cell culture medium was replaced with Opti-mem (Gibco) 6 hr post transfection. After 48 h the transfected cells were infected at different MOIs as indicated in Figures. At 17 hr p.i. samples from the supernatant were assayed for luciferase activity using the *Renilla* Luciferase Assay system (Promega) according to the manufacturer's instructions. The relative light units (RLU) were determined by GloMax Discover System GM3000 luminometer (Promega). To check effect on virus entry from knock-out of sialyltransferases, Sia-independent enterovirus strain RlucCVB3 genome replication was measured by intracellular *Renilla* luciferase activity. MOI = 0.1 of RlucCVB3 was used to infect HEK293 or HEK$^{\Delta Sia}$ cells. Infected HEK293 and HEK$^{\Delta Sia}$ cells were lysed at 8 hr p.i. and luciferase activity was measured following the manufacturer's protocol (*Renilla* luciferase assay system; Promega).

**Quantification of Sia gradients by Flow cytometry**. HEK$^{\Delta Sia}$ cells were seeded in 6-well plates (1.5E + 05 cells/well) and the next day cotransfected with a pEGFP reporter plasmid and different amounts and ratios of sialyltransferases (ST6Gal1 and/or ST3Gal4). After 48 h transfected cells were released using Cell Dissociation Buffer (Gibco), washed once in phosphate buffered saline (PBS), and fixed with 4% paraformaldehyde (PFA) in PBS. Cells firstly were incubated with biotinylated-lectins (MAL I, Vector Labs; SNA, Vector labs) for 1 hr and then complexed with Streptavidin for 1 hr, (Alexa Fluor™ 568 conjugate (1 mg/mL),Thermo Fisher). Cells were analyzed on BC Cytoflex LX (Beckman) using CytExpert fot CtytoFLEX Acquisition and Analysis software. The gating methods are based on standard protocols[67]. Representative flow cytometry gating strategy for lectin staining cells are shown in Supplementary Fig. 7.

**Software and statistical analysis**. All experiments were performed with $n = 3$, $n = 4$ or $n = 6$ biological replicates (as indicated). Each BLI binding assay was repeated at least twice, and binding signals of representative samples were the average of technical duplicates from the same sample. Virus initial binding rates, corresponding to the slopes of the binding curves during the first 5 min, were corrected by virus particle numbers and determined by Simple linear regression (GraphPad Prism 8.4.0). Statistical significance for bar diagram plots of virus binding or entry was determined using the unpaired t-test with Welch's correction, applying two-tailed calculation on biological replicates ($n$ is indicated in the figure legends) (GraphPad Prism 8.4.0). Fractional receptor densities correlating with half maximum initial binding rates were determined by non-linear regression analysis and performed by the Sigmoidal, 4PL, X is log (concentration) (GraphPad Prism 8.4.0). Each luciferase biological replicate is the average of technical triplicates from the same sample. Significance analysis of luciferase reporter assay was based on one way ANOVA followed by Tukey's multiple comparisons test (GraphPad Prism 8.4.0).

**Reporting summary**. Further information on research design is available in the Nature Research Reporting Summary linked to this article.

## Data availability
All data generated or analysed during this study are included in this published article (and its supplementary information files). Source data are provided with this paper.

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

## Acknowledgements

Support by a personal grant from the Chinese Scholarship Council to ML was obtained (201908350116). Support by the Lundbeck Foundation, Novo Nordisk Foundation, and the Danish National Research Foundation was obtained (DNRF107L: C.B., Y.N and H.C.). Recombinant virus strains VI75, BK79, WU95 and FU02 were kindly provided by Dr. R. Fouchier (Viroscience, Erasmus Medical Center, Rotterdam, the Netherlands). H5N8 was kindly provided by Wageningen Bioveterinary Research (Olaf de Leeuw, Ben Peeters and Riks Maas). Biotinylated synthetic glycans were kindly provided by Dr. Geert-Jan Boons (Utrecht University, Utrecht, the Netherlands). Figures 1g, 3a and 6 were created with BioRender.com

## Author contributions

M.L., C.A.H., and E.V. conceptualized and designed the experiments. M.L., L.Z.H., and A.A.S. performed the experiments. M.L., C.A.H., and E.V. wrote the manuscript with contributions from C.B., Y.N., F.J.K., and H.C. C.B., Y.N., and H.C. contributed essential materials and protocols. F.J.K., C.A.H., and E.V. supervised the project.

## Competing interests

The authors declare no competing interests.
