## [Peer Review File · Nature Communications]

Reviewers' comments:

Reviewer #1 (Remarks to the Author):

The manuscript entitled „ Human-type acid receptors contribute to avian influenza A virus binding and entry by hetero-multivalent interactions” by Liu and colleagues reports on the flexibility in sialoside recognition of HA from avian and human pathogenic influenza strains, which is of great relevance for the understanding of host adaptation. The authors obtained convincing data from experiments with biolayer interferometry, and to some extent from recombinant experiments with HEKdeltaSA cells. By using mixed sialic acid gradients, the authors were able to find enhancement binding to mixes of low amounts of “strong” receptors and increasing amounts of “weak” receptors, which gives strong evidence how avian viruses can evolve towards human hosts. I believe these findings are not only important for influenza virus research, but also for many other topics in the heterovalency field. The authors demonstrated impressively how BLI, as a powerful technique for binding characterization, can help to dissect complex heterovalent binding mechanisms. Therefore, this approach could motivate also other scientists (especially virologists) in studying host adaptation mechanisms. The article is well written, and to a large extent supported by references to the state-of-the-art in influenza virus research. I therefore highly recommend to consider this article for publication in nature communication, after revising several minor points:

1. In the introduction the kind of sialic acid (Neu5Ac or Neu5Gc) is not specified.

See also: DOI:<https://doi.org/10.1128/JVI.01567-19>

2. Phrase 66-68 can be a little bit misleading from my point of view. I assume it is meant “how mixes of non-preferred receptors can compensate for insufficient amounts of preferred receptors”. Otherwise, the phrase would describe from my perspective rather the general multivalency concept.

3. At the end of the introduction the evolutionary pathways towards host adaptation was mentioned. However, one could phrase it probably broader, by adding viral escape from antiviral decoy structures (see: <https://doi.org/10.1021/acs.jmedchem.1c00794>).

4. The authors claim, that that the dogma on either 2,3 SLN or 2,6 SLN specificity in IAV binding has not been convincingly weakened before. I would not fully agree here. The swine with higher ratios of 2,3/2,6 (compared to humans) as an intermediate host indicated the relevance of heterovalency or receptor flexibility since a long time. Further, histopathological studies with humans infected with avian flu showed already binding not only to 2,3 SLN displaying cells.

5. Sauter et al. (reference 18) described a secondary SA binding site in HA. Here, higher binding of 2,3 than 2,6 SLN was found, while the main RBS binds 2,6 SLN. The relevance of the secondary binding site in bivalent binding was demonstrated by Bandlow et al. 2017 (JACS). Although there no heterovalent binding was found, the relevance of a low affinity binding site could be enhanced by bridging to a higher affinity binding site.

Binding of 2,6 SLN to the primary binding site of avian flu HA is less likely, due to the fact of a smaller binding site (doi: [10.1002/embj.201387442](https://doi.org/10.1002/embj.201387442)).

6. The authors employed Oseltamivir carboxylate as NA inhibitors to probe only for HA interaction with the sialoside receptors. As the viral HA/NA is also bearing glycans, which is known to facilitate clustering of virus particles without NA activity, I was wondering whether a control of virus clustering within the experimental time was observed. Otherwise, an additional binding scenario (virus to virus) could occur on the BLI tip and maybe bias the experiments.

7. In Figure 1 it is not clear to me what “nm” on the y-axis means. How do the authors explain in 1 b that no saturation can be reached? By comparing 1c with 1 d, do the authors agree, that virus adaptation (by considering only the kind of sialoside receptors) is rather unidirectional from birds to humans, but not vice versa? The statistical analysis is not clear to me, as an N of 3 is not sufficient to perform a normality test as requirement for a t-test. Despite I think the biological effects are also convincing without statistical test in figure 1.

8. Throughout the whole manuscript the authors report on binding rates, but only show binding saturations. I could not find in the extended data information on the k_{on} or k_{off} rates, which could be informative (especially in order to justify the model in figure 6). I believe the authors refer in e.g. figure 2, 3 to “binding saturation in %” at the binding rate equilibria. If this is not the case, an explanation would be helpful.

9. In several figures the y-axis is only labeled with “binding (xy)”. It would be very helpful to add either “virus binding” or “lectin binding”, as it would simplify the reading. Further, adding the lectin specificity of either SLN or MAL 1 on the x-axis in fig. 3 d-g would also facilitate the understanding of the figures.

10. In line 152 figure 3e and 3f was referenced, but I believe 3d and 3e was meant.

11. In line 159 figure 3d-g was referenced, but I assume 3d-g was meant.
12. The authors state, that the Sia density correlates with the amount of transfected sialyltransferase. However, the FACS histograms in S Fig. 5 do not fully support this finding in the amount of expressed sialic acid. Especially for the figures S Figure 5 a-e, no visible shift in the Sia signal is detectable. Do the authors have an explanation for this finding? Further an explanation for the used colors in S Fig 5 a-j would be helpful.
13. In Figure 5 the statistical post test to assess significance is not mentioned (Bonferroni?)
14. From Figure 3 it did not become clear to me, what is meant with receptor density in the case of Lamp I. Sia density on Lamp I or Sia density on the BLI sensor tip?
15. I believe the model shown in Figure 6 is somewhat speculative, especially as the binding with an active NA with Sia binding and receptor elimination functions (which was not experimentally addressed by the authors). Figure 6 a is based on state-of-the-art research, but I don't see here additional findings from the study in hand. Although kinetic studies were conducted in this article k_{on} and k_{off} rates were not dissected, but that would have been very informative (even by providing the raw signals). Rather the equilibrium constant K_D was used 300s after the measurements as binding end points. Also the correlation of Sia receptor binding/clustering in cell signaling was not addressed by the authors, and is from my point of view very speculative at the moment. Therefore, I believe this model should be rather part of a review.

Reviewer #2 (Remarks to the Author):

It is well recognized that influenza receptor binding is not as clear-cut as is often described in the literature, with statements that human viruses bind to alpha2-6 and avian viruses bind to alpha2-3 linked sialic acid and a change in this specificity is required for a change in permissible host. The authors are querying this dogma and have tested binding and entry of influenza viruses on substrates and cells containing engineered mixtures of alpha2-6 and alpha2-3 linked sialic acids attached to glycans or glycoproteins. Interestingly, they find that a non-canonical, low-affinity receptor enhances binding and entry of virus; ie mixing alpha2-3 with the high-affinity alpha2-6 sialic acid increases binding and entry of human viruses and the converse for an avian virus. The results are consistent and the experiments well designed. The authors have attempted to find a mechanistic explanation for these results, but the model is not very convincing. They discuss the importance of multivalent binding but do not adequately explain how the "minority" receptor plays a role. The NA second binding site is not usually seen in human viruses, but maybe it becomes significant in this context. There was also evidence of a second binding site on HA (Sauter et al PNAS 1992) that might become significant under the experimental conditions used.

Additional comments

1. It would be useful to know if the viruses that provided the HA and NA for the recombinant viruses being studied were isolated in mammalian cells or eggs. Two of the viruses apparently contain mutations but there is no indication if these mutations affect the binding.
2. A key component in these studies is the viral neuraminidase. Although there are reports of the linkage specificity changing with a host change, the NA is known to always prefer the a2-3 sialylated substrates; the ratio may change but the activity is higher with alpha2-3. In the BLI experiments reported in this manuscript the NA active site is blocked by oseltamivir but it is unclear if this was used in every experiment. The avian N1 has an additional binding site that this group has previously studied and some discussion of its possible role would be useful here. Certainly some experiments without the NA inhibitor would be more insightful in terms of understanding viral attachment and entry.
3. Quantitation using the lectins MAL-1 and SNA is not convincing. For each, the signal may be linear with concentration, but how do the signals for 2-3 and 2-6 compare with each other?
4. It is unclear why LAMP-1 was chosen for these experiments. It is heavily glycosylated and its many glycans may not be fully processed when over-expressed.
5. Figure 1: Panel G is useful to visualize the experiment, but the second row only refers to panel C, not D, and similarly the next row is correct for panel E but not F.

6. Figure 2: the "Rate %" on the y axes is hard to interpret. It would be much clearer if actual rates were used as in Figure 3 b and c. In particular, is the maximum rate (here set to 100%) the same for both the red and black curves? If not, the conclusions might be different.
7. Figure 3: the y axis in panel E is labeled differently to D, F and G. Since it goes to 100, this might be an error in the label.
8. Figure 4: is there an explanation for the fall in entry after the maximum? How does increased sialyltransferase thereafter inhibit entry? Panel 4e is not clear to me – what does it show?
9. Figure 5: should be a-d, not e-h.
10. The model in Figure 6 is reasonably based on the concepts of multivalent binding but does not include the trimeric HA sites or possible binding at the NA second site or perhaps the putative second HA site where the lost esterase activity would reside.

Reviewer #3 (Remarks to the Author):

The study by Mengying Liu et al. reveals that non-binding human-type receptors efficiently enhance avian IAV binding and vice versa. The authors employ a panel of cleverly designed experimental systems, in which the ratio of human- to avian-type receptors can be varied, to show that heteromultivalent binding of IAV occurs. The conclusions are well supported by the data presented (with a few exceptions, see below) and thus, this manuscript provides strong evidence that our current view on IAV receptor binding needs to be revised. Given the importance of IAV receptor recognition for IAV biology and zoonotic potential of IAV these exciting findings are of high relevance for the fields of virology and cell biology.

Major points:

- While the authors use multiple human strains of IAV, they only use one avian strain. However, the authors generalize their observation to all avian strains (e.g. in the title). Additional avian strains should be included to allow for the conclusion that non-binding human-type receptors efficiently enhance avian IAV binding.
- Fig. 4e: From the shape of the black curve, the authors conclude that IAV binding on cells occurs in a super-selective manner. While this is a reasonable interpretation, the black curve alone is not proof for super-selective binding of IAV on cells. This needs further experimental evidence. One could add MAL I to regular cells (not the tunable HEK) or use specific sialidases before adding the virus and compare the slopes of the curves.
- Fig. 5: The error bars are rather high for the data sets that show heteromultivalent binding. Given the importance of this figure for the main findings of the manuscript, I suggest to add a third biological replicate and show the individual data points in the bar graph. This will enable the reader to assess the robustness of the findings better.
- Fig. 6: The role of NA is discussed in the figure legend and the main text but this is not reflected in the graphical model. I suggest that the authors also depict the function of NA in the model to highlight the important HA/NA interplay.

Minor points:

- Fig 3e Y axis label: Should this be "Binding rate (%)" as in 3d, 3f and 3g?
- Line 130: Fig 3 b and c should be referenced, not c and d.
- Supp. Fig. 1 is not referenced in the text.
- Supp. Fig. 3c: Which lectin was used in panel c? It is confusing that the colours in panel c correspond to different amounts of transfected plasmid whereas the same colours correspond to different lectins in a-b.
- Supp. Fig. 4a: In my opinion, it would be informative and important to also show HEK delta SIA cells in this graph to get an idea of the amount of sialic acid left after knockout and/or the levels of sialic acid independent entry. Fig. 3 gives some clues in this direction but it would be good to see the background levels in Supp. Fig. 4a.
- Supp. Fig. 4: The X axis is labeled with "Virus dilution". Should this not be "Relative virus concentration"? Furthermore, some numbers on the X axis labeling are in superscript.

- Supp. Fig 5: The authors should consider adding the amounts transfected and lectin used within the graph, to make it easier for the reader.
- L. 37-39 "IAV infection is initiated by binding of hemagglutinin (HA) on the viral envelope to epithelial cell surface-exposed Sia receptors." This is true for most strains of IAV but not for all as bat IAV of the H17 and H18 subtypes do not use sialic acid for host cell entry. The authors should specify, e.g. by saying "Human and avian IAV infection..." or include a statement about bat IAV.

Rebuttal to reviewers comments.

Comments of reviewers are included, followed by a **response** to every point. Changes made in the text are also inserted in the rebuttal in *"italics"* and their position in the manuscript is indicated according to the line numbering of the revised version (with track changes hidden).

Reviewer #1 (Remarks to the Author):

The manuscript entitled „ Human-type acid receptors contribute to avian influenza A virus binding and entry by hetero-multivalent interactions“ by Liu and colleagues reports on the flexibility in sialoside recognition of HA from avian and human pathogenic influenza strains, which is of great relevance for the understanding of host adaptation. The authors obtained convincing data from experiments with biolayer interferometry, and to some extent from recombinant experiments with HEKdeltaSA cells. By using mixed sialic acid gradients, the authors were able to find enhancement binding to mixes of low amounts of “strong” receptors and increasing amounts of “weak” receptors, which gives strong evidence how avian viruses can evolve towards human hosts. I believe these findings are not only important for influenza virus research, but also for many other topics in the heterovalency field. The authors demonstrated impressively how BLI, as a powerful technique for binding characterization, can help to dissect complex heterovalent binding mechanisms. Therefore, this approach could motivate also other scientists (especially virologists) in studying host adaptation mechanisms. The article is well written, and to a large extent supported by references to the state-of-the-art in influenza virus research. I therefore highly recommend to consider this article for publication in nature communication, after revising several minor points:

1. In the introduction the kind of sialic acid (Neu5Ac or Neu5Gc) is not specified.

Response: We have inserted this in line 35: *“which in avians and humans exclusively occur as the N-acetylneuraminic acid type”*

2. Phrase 66-68 can be a little bit misleading from my point of few. I assume it is meant “how mixes of non-preferred receptors can compensate for insufficient amounts of preferred receptors”. Otherwise, the phrase would describe from my perspective rather the general multivalency concept.”

Response: We clarified line 69-72 by modifying this sentence according to the reviewers’ suggestion: *“We showed that, by heteromultivalent binding, synthetic sialoglycans which by themselves cannot support binding can compensate for insufficient amounts of preferred receptors for avian as well as human IAV strains.”*

3. At the end of the introduction the evolutionary pathways towards host adaption was mentioned. However, one could phrase it probably broader, by adding viral escape from antiviral decoy structures (see: <https://doi.org/10.1021/acs.jmedchem.1c00794>).

Response: We think that the reviewer refers to the last paragraph of the discussion as the end of the introduction does not deal with this subject. We fully agree that an important aspect of adaptation is the need of a virus to deal efficiently with decoy receptors. Natural decoy receptors like mucins differ between hosts and viruses need to adapt to this when changing host. We modified the last sentence of the discussion to incorporate this point dealing with these issues (lines 331-333): *“Hetero-multivalent interactions with such areas on the respiratory epithelium, as well as with decoy receptors present in the overlaying mucus, could be a key factor in the evolutionary pathways leading to host adaptation.”*

4. The authors claim, that that the dogma on either 2,3 SLN or 2,6 SLN specificity in IAV binding has not been convincingly weakened before. I would not fully agree here. The swine with higher ratios of 2,3/2,6 (compared to humans) as an intermediate host indicated the relevance of heterovalency or receptor flexibility since a long time. Further, histopathological studies with humans infected with avian flu showed already binding not only to 2,3 SLN displaying cells.

Response: The dichotomy of the predominant binding of avian IAVs to 2-3Sia receptors and human IAVs to 2-6Sia receptors [citations 4,5] has been consistently confirmed for the large majority of new avian or human isolates that were tested until now, mostly employing platforms homogeneously coated with either 2-3 or 2-6Sias [citations 13,26,27]. It is the common perception that receptors identified as non-binders in such assays do not contribute to binding and infection [citations 9, 10] that is challenged here. A heterogenous receptor repertoire, including a diverse distribution of 2-3Sias and 2-6Sias has been found in many host species as shown by histopathological studies as well as by detailed glycomic analyses of different tissues from swine, human, and avian species [citations 8,35-37]. Such environments can support entry of viruses preferring different linkage types [citations 8, 11-14] but it is usually concluded that binding and entry is supported by the high-affinity receptors present in such a receptor landscape. We have modified lines 43-45 to incorporate this point more clearly: "*Often, α 2-3 and α 2-6 Sia linkages are found coinciding, at various ratios, on the different types of glycoproteins and glycolipids found at these sites which is thought to support the occasional infection of specific stretches of the swine or human respiratory tract by some avian IAVs.*"

5a. Sauter et al. (reference 18) described a secondary SA binding site in HA. Here, higher binding of 2,3 than 2,6 SLN was found, while the main RBS binds 2,6 SLN. The relevance of the secondary binding site in bivalent binding was demonstrated by Bandlow et al. 2017 (JACS). Although there no heterovalent binding was found, the relevance of a low affinity binding site could be enhanced by bridging to a higher affinity binding site.

Response: A secondary binding site on HA has indeed been described. Although it has not been investigated to which extent a functional secondary binding site is present in different IAV strains and genotypes it could have an effect on (heteromultivalent) binding. Reviewer 2 (point 10) also suggests to discuss this issue and reviewers 2 and 3 suggest to also include the NA secondary receptor binding site and NA active site in the model. To address the remarks of all three reviewers on potential effects on heteromultivalent binding of secondary binding sites that were identified on HA and NA as well as on the activity of NA we have replaced panel B by two panels that illustrate their potential contribution as described by lines 256-265 in the discussion: "*Multiple reports have described the presence of a secondary sialic acid binding site (2SBS) on NA [38,39] as well as a secondary sialic acid binding site, called the vestigial esterase subdomain (VES), on HA [18]. The 2SBS is conserved in avian, but not in human IAVs, [40-42] and displays 2-3Sia specificity although a 2SBS of much lower affinity was suggested to be present in some pandemic H1N1 strains [43]. Binding affinity for the putative VES was estimated to be \sim 200mM [44]. Such potential binding sites of low affinity may expand the options of forming different heteromultivalent interaction networks as illustrated in Fig. 6b and is further described in the legend to Fig. 6b. These diverse and highly dynamic interaction networks may in turn affect the efficiency of NA to associate with, and cleave, Sias, thereby influencing virus motility and dissociation (Fig 6c).*" The legend was also adapted to describe the changes.

5b. Binding of 2,6 SLN to the primary binding site of avian flu HA is less likely, due to the fact of a smaller binding site (doi: 10.1002/embj.201387442).

Response: The somewhat narrower binding site of avian IAVs might be one of the determining factors leading to a higher K_D for 2-6Sias. Still, 2-6Sias do bind to the primary receptor binding site of avian IAVs as shown by the co-crystallization of 2-6Sia in complex with avian HAs as well as by the determination of a K_D for 2-6Sia interaction with avian HAs by a number of different methods that was shown to be affected by mutations in the primary receptor binding site.

6. The authors employed Oseltamivir carboxylate as NA inhibitors to probe only for HA interaction with the sialoside receptors. As the viral HA/NA is also bearing glycans, which is known to facilitate clustering of virus particles without NA activity, I was wondering whether a control of virus clustering within the experimental time was observed. Otherwise, an additional binding scenario (virus to virus) could occur on the BLI tip and maybe bias the experiments.

Response: All virus strains were grown in MDCK cells under standard conditions in absence of oseltamivir. The activity of NA removes all sialic acids from HA as well as from NA and virus so virus binding will therefore not occur.

7a. In Figure 1 it is not clear to me what “nm” on the y-axis means. How do the authors explain in 1 b that no saturation can be reached? By comparing 1c with 1 d, do the authors agree, that virus adaptation (by considering only the kind of sialoside receptors) is rather unidirectional from birds to humans, but not vice versa?

Response: Panels a-f display the primary binding curves obtained by BLI. Binding is detected by a change in light reflection which is detected by the optical sensor and by the algorithms used by the system expressed as a shift in nanometers. The y-axis thus expresses the absolute level of binding with nm as a unit. For a detailed account see citation 23. The curves do reach saturation in time (approximately around 10 nm) but in Fig 1 binding was only performed for 10 min. Binding rate is initially stable but starts to decline slightly in time (as seen by slight bending of the curves). Panel c shows that binding of human H3N2 to a low concentration of human-type 2-6Sia receptor is assisted by avian-type 2-3Sia receptor. These receptors both occur in the human respiratory tract, and therefore heteromultivalent binding could occur and have an effect on infection and evolution within the human host as well as on the motility on a specific cell (all mentioned in the discussion). Indeed human IAVs are not expected to evolve into avian IAVs although that cannot be concluded from panels c and d which just show the occurrence of heteromultivalent interactions for human and avian IAVs.

7b. The statistical analysis is not clear to me, as an N of 3 is not sufficient to perform a normality test as requirement for a t-test. Despite I think the biological effects are also convincing without statistical test in figure 1.

Response: The unpaired t-test with Welch’s correction was used, applying two-tailed calculation from within the graphpad prism package. This description was added to the methods section (lines 450 to 452): *“Statistical significance for bar diagram plots of virus binding or entry was determined using the unpaired t-test with Welch’s correction, applying two-tailed calculation on biological replicates (n is indicated in the figure legends) (GraphPad Prism 8.4.0)”*

8a. Throughout the whole manuscript the authors report on binding rates, but only show binding saturations.

Response: We have not shown binding saturation in the figures, only figure 1 shows primary binding curves but binding does not proceed to saturation there. We have determined the initial binding rates from the initial slope of the primary BLI virus binding curves as described in the methods section (lines 406-409): *“Virus initial binding rates were calculated from virus binding levels after 5 minutes (during which the binding curve is linear apart from the first few seconds). Initial binding rates were either plotted absolute (nm/10⁹ virus particles) or relative to the maximum binding rate at full receptor density as indicated in the Figures.”* In reference 23 it was previously discussed why the initial binding rate is considered to be the most important binding parameter for comparing virus strains. To avoid any confusion we have adapted the text as follows (Line 95-98): *“Virus binding rates were determined from the slope of the binding curves (examples shown in Supplementary Figure. 1a,b). Binding rates were normalized to the highest binding rate for each virus and plotted against the relative density of the high-affinity receptor (Fig. 2a)”*.

8b. I could not find in the extended data information on the k_{on} or k_{off} rates, which could be informative (especially in order to justify the model in figure 6). I believe the authors refer in e.g. figure 2, 3 to “binding saturation in %” at the binding rate equilibria. If this is not the case, an explanation would be helpful.

Response: We agree that the k_{on} and k_{off} of individual HA-receptor interactions are important parameters. They determine (in part) virus binding rate, virus mobility and the effects of heteromultivalent binding as discussed in the model. They are in the mM range which has hampered a systematical analysis of these parameters for larger sets of IAV strains. In contrast, the K_D for the “virtually irreversible” binding of an IAV particle to a receptor surface is in an

extremely low range (pM to fM) that prevents accurate determination by equilibrium binding methods using techniques like BLI or SPR and is also discouraged by the manufacturers. As explained under 8a we do not refer to binding saturation but to binding rate.

9. In several figures the y-axis is only labeled with "binding (xy)". It would be very helpful to add either "virus binding" or "lectin binding", as it would simplify the reading. Further, adding the lectin specificity of either SLN or MAL 1 on the x-axis in fig. 3 d-g would also facilitate the understanding of the figures.

Response: We welcome the suggestions on improving the clarity of the figure axes and have made the following modifications:

Fig 1: Virus binding (y-axis) and virus binding rate (%) (y-axis) of inserted bar diagrams.

Fig 2: Virus binding rate (y-axis)

Fig 3b,c: Virus binding rate (nm/300s.10⁹ particles) (y-axis)

Fig 3d-g: Virus binding rate (%) (y-axis)

Fig 4e: Virus or SNA binding (%); Virus entry (%) (y-axis)

Suppl Fig 1: Virus binding (%) (y-axis)

Suppl Fig 2: Virus binding rate (%) (y-axis)

Suppl Fig 3: Virus binding rate (%) (y-axis)

Suppl Fig 4a,b and d: Lectin binding (nm) (y-axis)

Suppl Fig 4c: SNA binding (nm) (y-axis)

10. In line 152 figure 3e and 3f was referenced, but I believe 3d and 3e was meant.

Response: The reviewer is right, we have corrected these errors.

11. In line 159 figure 3d-g was referenced, but I assume 3d-g was meant.

Response: The reviewer is right, we have corrected these errors.

12. The authors state, that the Sia density correlates with the amount of transfected sialyltransferase. However, the FACS histograms in S Fig. 5 do not fully support this finding in the amount of expressed sialic acid. Especially for the figures S Figure 5 a-e, no visible shift in the Sia signal is detectable. Do the authors have an explanation for this finding? Further an explanation for the used colors in S Fig 5 a-j would be helpful.

Response: The reviewer is right that the gradual shift of the peaks in response to increased amounts of transfected sialyltransferase is hard to see by comparing the individual FACS histograms. We therefore replaced the individual histograms of Supplementary Fig 5 by an overlay of the histograms to showing more clearly the shifts. The reason for the limited shift is the high background signal of SNA or MAL I binding to mock-transfected cells. This is a well-known property of these lectins and therefore they are seldomly used for quantitative analysis by FACS but mostly for a qualitative/semi-quantitative analysis by fluorescence microscopy. We followed the reviewers' suggestion in improving the labeling of the figure and added the geometric mean of each histogram to the figure. The legend to Supplementary Fig 6 (original Supplementary Fig 5) was adapted as follows (supplementary lines 67-72): "*FACS analysis of HEKΔSia cells transfected with ST3Gal4 (2-3Sia) (a) or ST6Gal1 (2-6Sia) (b) at a concentration range from 0 to 90ng as shown in the color-coded table. Cells were stained (Streptavidin-Alexa568) by incubation with biotinylated lectins MAL I (a) or SNA (b). Cells were co-transfected with a GFP expression plasmid and GFP+ cells were gated (488nm). X-axis displays fluorescent emission by the 584-42 (561) channel. Y-axis is cell count.*"

13. In Figure 5 the statistical post test to assess significance is not mentioned (Bonferroni?)

Response: The unpaired t-test with Welch's correction was used, applying two-tailed calculation from within the graphpad prism package. This description was added to the methods section (lines 450 to 452): "*Statistical significance for bar diagram plots of virus binding or entry was determined using the unpaired t-test with Welch's correction, applying two-tailed calculation on biological replicates (n is indicated in the figure legends) (GraphPad Prism 8.4.0)*".

14. From Figure 3 it did not become clear to me, what is meant with receptor density in the case of Lamp I. Sia density on Lamp I or Sia density on the BLI sensor tip?

Response: We can understand the confusion but the answer is: both, with different densities on the BLI sensor resulting from different densities on LAMP I. Different Sia densities on LAMP I were generated by co-transfecting Lamp I with increasing concentrations of sialyltransferases. Differentially sialylated LAMP I proteins were then loaded to maximal density on the biosensors and the total SIA-content on the sensor was subsequently analysed by lectin binding as shown in Supplementary figure 3. This leads us to the conclusion (lines 147-150) that: "*Sia density of complementary 2-6Sia/2-3Sia gradients expressed on LAMP I can be quantified by lectin staining. Therefore, plotting lectin staining against virus binding will allow to identify hetero-multivalent binding effects for IAV binding to natural glycoprotein receptors.*" Thus, the relative SNA or MAL I binding levels plotted on the x-axis are a measure for the SIA density on the sensor fully loaded with the different LAMP I proteins that carry different amounts of receptor (every point represents a sensor loaded with another LAMP I protein, illustrated in Fig 3a).

15a. I believe the model shown in Figure 6 is somewhat speculative, especially as the binding with an active NA with Sia binding and receptor elimination functions (which was not experimentally addressed by the authors).

Response: We actually did test the effect of heteromultivalent interactions in the absence of the NA inhibitor OC. The entry studies (Fig 5) displayed highly significant effects of a heterogeneous receptor surface implying that the effects shown in presence of OC in the binding assays must also be active in absence of OC. We modified the description of results to better show this point (lines 219-222): "*Importantly, all entry experiments were performed in the absence of the NA inhibitor OC. Therefore, we conclude that heteromultivalent binding effects enhance entry efficiency of IAV and can be observed under natural conditions where NA is active.*"

We do think that we added essential information on heteromultivalent binding in relation to entry by the experiments shown in Fig 5. However, in view of the reviewers comments we have removed the entry model (panel B) and replaced this by panels illustrating the secondary binding sites of HA and NA as well as the NA active site in response to the other two reviewers. We reduced the discussion on entry which now reads as (lines 287-301): "*Receptor-bound IAV particles are highly motile [22,47] and explore the cell surface for minutes before entry [48,49]. This NA activity-driven directional motility and entry efficiency depends on balanced NA and HA activity [23,47,50]. NA activity may be affected by hetero-multivalent interactions as, like HA, NA displays receptor specificity. Avian IAVs, engaging 2-3 Sias for primary attachment, preferentially cleave 2-3Sias over 2-6Sias (Fig 6c, $k_{cat1} > k_{cat2}$). Upon primary attachment to a heterogeneous receptor surface via high-affinity receptors, migration to poorly cleavable 2-6Sia-rich clusters could reduce motility which has been observed to precede cell entry [48,49]. IAV particle binding-induced clustering of sialylated receptors at such spots [51] is signaling induction of their uptake by endocytosis [34,48,51-56]. High-avidity binding via low-affinity interactions has been shown to efficiently induce clustering and signaling in other systems [57]. Thus, signaling receptors decorated with Sias with either low or high affinity could be clustered by any particular IAV strain translocating over the cell surface upon binding initiated by hetero-multivalent interactions at a distant spot. In summary, entry efficiency can directly be enhanced through hetero-multivalent interactions, increasing binding rate and possibly exerting effects on virus motility, receptor clustering and signaling.*"

15b. Figure 6 a is based on state-of-the-art research, but I don't see here additional findings from the study in hand. Although kinetic studies were conducted in this article k_{on} and k_{off} rates were not dissected, but that would have been very informative (even by providing the raw signals). Rather the equilibrium constant K_D was used 300s after the measurements as binding end points.

Response: We think that without visualizing the model as in Fig 6a the discussion on the mechanism by which heteromultivalent binding works and impacts virus-cell surface interactions will be hard to understand. As discussed under point 8b we did not (and cannot) determine an equilibrium K_D but the initial binding rate (the slope of the primary binding curve). Under point 8 we also discussed that the k_{on} and k_{off} of virus particles cannot be derived from the raw signals as

the extremely tight virus binding results in primary binding curves that do not fit to equilibrium binding models because of the extremely low K_D of virus particles. The determining parameters for virus binding rate, virus mobility and the effects of heteromultivalent binding are the k_{on} and k_{off} of individual HA-receptor interactions. The K_D for these interactions is in the mM range, complicating methods for its determination. This was described in the introduction and discussion but to highlight this crucial information we have modified a section in the legend to the model (line 718-723): *"The initial, intermolecular and virus concentration dependent, monovalent interaction between a 2-3Sia and an HA on the virus surface ($K_D=k_{off}/k_{on} \sim 1-20$ mM) is characterized by a low binding rate constant k_{on} ($M^{-1} s^{-1}$, formation of interactions is slow) and a high dissociation rate constant k_{off} (s^{-1} , interactions are short-lived with a half-life time of 0.5 to a few seconds) [17-21]. Additional interactions (multivalent binding) will lead to longer lasting virus-to-surface binding that is detected by BLI."*

15c. Also the correlation of Sia receptor binding/clustering in cell signaling was not addressed by the authors, and is from my point of view very speculative at the moment. Therefore, I believe this model should be rather part of a review.

Response: We think it is valid to discuss the impact of novel findings, especially those on entry (Fig 4) in the context of previous findings on entry mechanisms. However we can agree with removal of panel B and restrict the discussion on this matter by shortening the text as outlined above under point 15a.

Reviewer #2 (Remarks to the Author):

It is well recognized that influenza receptor binding is not as clear-cut as is often described in the literature, with statements that human viruses bind to alpha2-6 and avian viruses bind to alpha2-3 linked sialic acid and a change in this specificity is required for a change in permissible host. The authors are querying this dogma and have tested binding and entry of influenza viruses on substrates and cells containing engineered mixtures of alpha2-6 and alpha2-3 linked sialic acids attached to glycans or glycoproteins. Interestingly, they find that a non-canonical, low-affinity receptor enhances binding and entry of virus; ie mixing alpha2-3 with the high-affinity alpha2-6 sialic acid increases binding and entry of human viruses and the converse for an avian virus. The results are consistent and the experiments well designed. The authors have attempted to find a mechanistic explanation for these results, but the model is not very convincing. They discuss the importance of multivalent binding but do not adequately explain how the "minority" receptor plays a role. The NA second binding site is not usually seen in human viruses, but maybe it becomes significant in this context. There was also evidence of a second binding site on HA (Sauter et al PNAS 1992) that might become significant under the experimental conditions used.

Response: We have adapted the explanation of the role of the "minority" receptor and have more specifically described the panels in the main text to avoid confusion (lines 237-251): *"We propose a model for heteromultivalent binding as shown in Fig. 6a. Because of the low affinity of HA-Sia interactions (K_D is ~ 1 to ~ 20 mM [18-20]), even the initial monovalent interaction between a viral HA and its preferred receptor (left panel, receptor-associated HAs are indicated in red) is very short-lived (half-life less than a few seconds). As a result, the balance is towards dissociation (black arrows). Still, some particles will form a second interaction, resulting in bivalent binding. The K_D ($=k_{off}/k_{on}$) of the initial interaction, as well as receptor density, are major factors that determine the frequency of bi-valent complex formation which, by its much higher avidity, promotes engagement of additional receptors. The equilibrium of an initial interaction with a non-preferred receptor of relatively low affinity (right panel) will however be further to the unbound state, thereby almost completely preventing a transition to bivalent binding. In contrast, at a surface of mixed low and high affinity receptors (middle panel), low affinity receptors can function in the formation of the second and subsequent interactions leading to increase of avidity by heteromultivalent binding, even though they are not able to initiate high-avidity virus binding by themselves (further details can be found in the legend to Fig. 6)"*

We further adjusted the model by replacing panel 6b (in response to reviewer 1) by which we discuss the potential effects of secondary receptor binding sites on HA and NA as described below under point 10.

Additional comments

1. It would be useful to know if the viruses that provided the HA and NA for the recombinant viruses being studied were isolated in mammalian cells or eggs. Two of the viruses apparently contain mutations but there is no indication if these mutations affect the binding.

Response: The recombinant viruses were isolated from cells as described by Koel et al [citation 56]. The HA and NA of BK79 and WU95 were derived from isolate A/Netherlands/233/82 and A/Netherlands/178/95 respectively. Both isolates differed at one position from the majority consensus sequence around that time and were therefore engineered at those positions to be similar to the majority consensus as described in citation 56. To avoid any confusion we have added the database accession numbers of the engineered genes to the methods section (line 337-349).

2. A key component in these studies is the viral neuraminidase. Although there are reports of the linkage specificity changing with a host change, the NA is known to always prefer the α 2-3 sialylated substrates; the ratio may change but the activity is higher with α 2-3. In the BLI experiments reported in this manuscript the NA active site is blocked by oseltamivir but it is unclear if this was used in every experiment. The avian N1 has an additional binding site that this group has previously studied and some discussion of its possible role would be useful here. Certainly some experiments without the NA inhibitor would be more insightful in terms of understanding viral attachment and entry.

Response: Oseltamivir carboxylate was used as an inhibitor in all BLI experiments (line 404). In the absence of an NA inhibitor, the BLI curves would represent the combination of virus binding events and virus release events, the latter increasing over time due to accumulating receptor depletion by NA activity. Such curves depend on the balance between HA and NA and associated virus motility [citations 23, 38]. The reviewer is right that the contribution of NA activity to heteromultivalent binding as determined by BLI is interesting but this requires extensive and novel experimental approaches beyond the scope of this paper in order to disentangle the contributions of HA and NA. Importantly, however, the demonstration of a multivalency effect on virus entry, which is performed in presence of an active NA, shows that the principle of heteromultivalent interaction also occurs under natural conditions.

The role of secondary receptor binding sites in NA and HA was also questioned by the other reviewers and therefore discussed according to a panel that we added to the model Fig 6 as described in the discussion (lines 256-265): *"Multiple reports have described the presence of a secondary sialic acid binding site (2SBS) on NA [201datab plus others] as well as a secondary sialic acid binding site, called the vestigial esterase subdomain (VES), on HA [18]. The 2SBS is conserved in avian IAVs [ref] and displays 2-3Sia specificity but a 2SBS of even lower affinity was suggested to be present in pandemic H1N1 strains [203datab]. Binding affinity for the putative VES was estimated to be \sim 200mM [464datab]. Such potential binding sites of low affinity may expand the options of forming different heteromultivalent interaction networks as illustrated in Fig. 6b and is further described in the legend to Fig. 6b. These diverse and highly dynamic interaction networks may in turn affect the efficiency of NA to associate with, and cleave, Sias, thereby influencing virus motility and dissociation (Fig 6c)"*

The legend was also adapted to describe the changes.

3. Quantitation using the lectins MAL-1 and SNA is not convincing. For each, the signal may be linear with concentration, but how do the signals for 2-3 and 2-6 compare with each other?

Response: In supplementary Fig. 4 we did show that LAMP I expression in combination with a concentration range of ST6Gal1 (Fig. S4a) or ST3Gal4 (Fig. S4b) gives a linear SNA (Fig. S4a, magenta line) or MAL-I (Fig. S4b, black line) binding curve (logarithmic x-axis) for the expressed

protein over a concentration range that was used to determine the heteromultivalency effects on binding and entry. To show more clearly that the SNA and MALI show highly similar slopes these lines were combined in the new panel Fig.S4d as now described in the legend (Supplementary lines 47 to 49): "(d) displays the SNA binding curve of panel a and the MAL I binding curve of panel b in a single diagram to show that increasing ST6Gal1 or ST3Gal4 give a similar quantitative effect on lectin binding."

In addition, Fig. 4e shows that logarithmic plots of SNA binding to LAMP I (ST6Gal1 range) or 2-6S(LN)2 (Sia concentration range of known density as loading is quantified by BLI) display a similar slope, which further validates the use of lectins to quantify Sia density.

4. It is unclear why LAMP-1 was chosen for these experiments. It is heavily glycosylated and its many glycans may not be fully processed when over-expressed.

Response: LAMP I was selected because it contains 18 N-linked glycosylation sites. We previously used fetuin (3 glycosylation sites; citation 23) but this protein supported lower virus binding rates. Moreover, the presence of large number of glycosylation provides much better options for expressing a gradient of sialic acids on a single protein molecule as required for the current project. We agree that, overexpression of a protein could lead to a reduction in glycan maturation and capping by sialic acids and we do not know the heterogeneity in glycosylation between individual LAMP I molecules. However, a comparison of the SNA binding curves to a gradient of synthetic sialoglycans and to LAMP I from cells transfected with different amounts of ST6Gal1 (Fig 4e) suggests similar Sia gradients as the curves overlap very well.

5. Figure 1: Panel G is useful to visualize the experiment, but the second row only refers to panel C, not D, and similarly the next row is correct for panel E but not F.

Response: The reviewer is right, we have adapted the figure to indicate that the panels refer to fig 1c and 1 e respectively.

6. Figure 2: the "Rate %" on the y axes is hard to interpret. It would be much clearer if actual rates were used as in Figure 3 b and c. In particular, is the maximum rate (here set to 100%) the same for both the red and black curves? If not, the conclusions might be different.

Response: For Figure 2a-f and Fig 3d-g we used the same way of plotting receptor density (x-axis) against relative binding rate (y-axis) as this is the easiest way to directly compare the curves for different virus strains. In Fig 3 we added panel b and c in order to compare the absolute binding rates to 100% density of 2-3S(LN)2 or 2-6S(LN)2 (panel 3c) to 100% density of 2-3SIA-LAMP I or 2-6SIA-LAMP I (panel 3b). From panel 3c it can be seen that the maximum binding rate for the different virus strains to sialoglycans maximally differs ~1.8-fold (between WU95 and FU02). We have added a sentence to the legend of Fig 2, referring to panel 3c, to better clarify the presence of this information (**lines 661-662**): "*The absolute binding rates of the different virus strains are plotted in Fig 3c showing that they differ less than 2-fold.*"

7. Figure 3: the y axis in panel E is labeled differently to D. F and G. Since it goes to 100, this might be an error in the label.

Response: This is indeed an error, we have modified it to be the same as for panel d, f and g.

8. Figure 4: is there an explanation for the fall in entry after the maximum? How does increased sialyltransferase thereafter inhibit entry? Panel 4e is not clear to me – what does it show?

Response: The decline in entry efficiency at high doses of transfected sialyltransferase is indeed a consistent and interesting phenomenon. We added the, to our opinion, most plausible explanation to the results section (lines 186-190) as it would disrupt the flow of the discussion when inserted in the latter section: "*Likely, this relates to other factors known to affect virus binding efficiency, like for instance the number of LacNAc repeats of a sialylated glycan antennae [16]. Capping of glycan chains by sialic acid will block the addition of additional sugar moieties and therefore glycan chain length is dependent on the level of sialyltransferase activity.*"

Panel 4e visualizes a comparison of the results displayed in Figures 2 to 4. The effects of receptor density on virus and lectin binding to synthetic glycans and glycoproteins, as well as on virus

entry, are shown side by side in a single panel by plotting them on similar scales. We modified the text to explain this more clearly: (lines 191 to 196): "To directly compare how receptor density affects virus binding and entry we re-plotted the binding and entry results for BK79 obtained above in parallel on the same scale (Fig 4e). BK79 entry efficiency (Fig 4e, black line) and BK79 binding rate to LAMP I (Fig.4e, solid red line) are plotted against ST6Gal1 on the lower x-axis and compared to BL79 binding to 2-6S(LN)2 (dotted red line) plotted against receptor density on an identical logarithmic scale (upper x-axis).".

9. Figure 5: should be a-d, not e-h.

Response: We repaired the error.

10. The model in Figure 6 is reasonably based on the concepts of multivalent binding but does not include the trimeric HA sites or possible binding at the NA second site or perhaps the putative second HA site where the lost esterase activity would reside.

Response: To address the remarks of all three reviewers on potential effects on heteromultivalent binding of secondary binding sites that were identified on HA and NA as well as on the activity of NA we have replaced panel B by two panels that illustrate their potential contribution as described by lines 256-265 in the discussion: "Multiple reports have described the presence of a secondary sialic acid binding site (2SBS) on NA [38,39] as well as a secondary sialic acid binding site, called the vestigial esterase subdomain (VES), on HA [18]. The 2SBS is conserved in avian, but not in human IAVs, [40-42] and displays 2-3Sia specificity although a 2SBS of much lower affinity was suggested to be present in some pandemic H1N1 strains [43]. Binding affinity for the putative VES was estimated to be ~200mM [44]. Such potential binding sites of low affinity may expand the options of forming different hetero-multivalent interaction networks as illustrated in Fig. 6b and is further described in the legend to Fig. 6b. These diverse and highly dynamic interaction networks may in turn affect the efficiency of NA to associate with, and cleave, Sias, thereby influencing virus motility and dissociation (Fig 6c). " The legend was also adapted to describe the changes."

Reviewer #3 (Remarks to the Author):

The study by Mengying Liu et al. reveals that non-binding human-type receptors efficiently enhance avian IAV binding and vice versa. The authors employ a panel of cleverly designed experimental systems, in which the ratio of human- to avian-type receptors can be varied, to show that heteromultivalent binding of IAV occurs. The conclusions are well supported by the data presented (with a few exceptions, see below) and thus, this manuscript provides strong evidence that our current view on IAV receptor binding needs to be revised. Given the importance of IAV receptor recognition for IAV biology and zoonotic potential of IAV these exciting findings are of high relevance for the fields of virology and cell biology.

Major points:

1. While the authors use multiple human strains of IAV, they only use one avian strain. However, the authors generalize their observation to all avian strains (e.g. in the title). Additional avian strains should be included to allow for the conclusion that non-binding human-type receptors efficiently enhance avian IAV binding.

Response: We agree that, considering the title, additional strains should be included to allow generalization of the principle of heteromultivalent binding. We have therefore confirmed the heteromultivalent binding principle for an avian H5N8 strain belonging clade 2.3.4.4 that has abundantly spread over the world since 2014. In addition we now also show heteromultivalent binding for the HU02 strain that lacks a glycan attached to position 158 in the receptor binding site showing that the presence or absence of this glycan, which is frequently found in H5 strains, does not affect heteromultivalent binding. Both panels are shown in supplementary fig. 2 and the text is modified as follows (lines 105 to 107): "To confirm that hetero-multivalent binding can have a positive effect on binding of other avian IAVs we show similar effects on an H5N8 virus strain and on another H5N1 IAV (Supplementary Fig. 2)".

2. Fig. 4e: From the shape of the black curve, the authors conclude that IAV binding on cells occurs in a super-selective manner. While this is a reasonable interpretation, the black curve alone is not proof for super-selective binding of IAV on cells. This needs further experimental evidence. One could add MAL I to regular cells (not the tunable HEK) or use specific sialidases before adding the virus and compare the slopes of the curves.

Response: We agree that solid proof requires more extensive experiments and have modified the text at three places where this was mentioned: (lines 203 to 206) "*Entry into cells (Fig 4e, black line) is detected at lower ST6Gal1 doses than binding to LAMP I. Super-selective virus binding translates into efficient virus entry above the receptor threshold as the slope of the BK79 entry curve is only slightly less steep than for BK79 binding*"; (lines 272-274) "*Strikingly, virus entry also rapidly increases above a threshold receptor density (Fig. 4e) suggesting that, in contrast to what has been frequently reported [8,11-14], there is a strong positive correlation between binding avidity and entry.*"; (line 697) "*Comparison of virus binding and entry kinetics*".

We did some initial attempts to assess heteromultivalency effects on entry into wildtype cells but encountered major technical difficulties. High levels of SNA or MAL I kill cells in short time and sialidases were not able to completely remove all sialic acids while quantification and characterization of the remaining sialic acids appears complicated. This clearly requires development of methods beyond the scope of this paper.

3. Fig. 5: The error bars are rather high for the data sets that show heteromultivalent binding. Given the importance of this figure for the main findings of the manuscript, I suggest to add a third biological replicate and show the individual data points in the bar graph. This will enable the reader to assess the robustness of the findings better.

Response: We performed a third biological replicate and added the results to Fig. 5. We have followed the suggestion of the reviewer to include the datapoints in the plot. Whereas a third replicate improved the significance, it hardly affected standard deviations and we therefore assume that this represents the spread that occurs in these complicated experiments with living cells.

4. Fig. 6: The role of NA is discussed in the figure legend and the main text but this is not reflected in the graphical model. I suggest that the authors also depict the function of NA in the model to highlight the important HA/NA interplay.

Response: To address the remarks of all three reviewers on potential effects on heteromultivalent binding of secondary binding sites that were identified on HA and NA as well as on the activity of NA we have replaced panel B by novel panels B and C. Panel C depicts the function of NA and is described in the discussion. The part on NA now shows as (lines 287-294): "*Receptor-bound IAV particles are highly motile [22,47] and explore the cell surface for minutes before entry [48,49]. This NA activity-driven directional motility and entry efficiency depends on balanced NA and HA activity [23,47,50]. NA activity may be affected by hetero-multivalent interactions as, like HA, NA displays receptor specificity. Avian IAVs, engaging 2-3 Sias for primary attachment, preferentially cleave 2-3Sias over 2-6Sias (Fig 6c, $k_{cat1} > k_{cat2}$). Upon primary attachment to a heterogeneous receptor surface via high-affinity receptors, migration to poorly cleavable 2-6Sia-rich clusters could reduce motility which has been observed to precede cell entry [48,49].*"

The legend to Fig 6c shows (lines 741-745): "*c, NA activity effects on heterogeneous receptor surfaces. NA catalytic site (white) activity depends on substrate structure and is for instance higher on 2-3Sia (k_{cat1}) than on 2-6Sia (k_{cat2}). Catalytic activity can be affected by arrangement of receptors via the different binding sites, for instance by binding to 2SBS (6). NA activity drives virus motility by reducing receptor density (7).*"

Minor points:

5. Fig 3e Y axis label: Should this be "Binding rate (%)" as in 3d, 3f and 3g?

Response: Yes, that was an error. We corrected the figure.

6. Line 130: Fig 3 b and c should be referenced, not c and d.

Response: Yes, another error, we corrected this.

7. Supp. Fig. 1 is not referenced in the text.

Response: It was referenced in the legend to Fig. 2 to explain how the plots were made. We now also referenced it in the text (line 96).

8. Supp. Fig. 3c: Which lectin was used in panel c? It is confusing that the colors in panel c correspond to different amounts of transfected plasmid whereas the same colors correspond to different lectins in a-b.

Response: SNA was used. We have adapted the colors of the legend of Fig S4c (original Fig. S3c) to avoid this confusion.

9. Supp. Fig. 4a: In my opinion, it would be informative and important to also show HEK delta SIA cells in this graph to get an idea of the amount of sialic acid left after knockout and/or the levels of sialic acid independent entry. Fig. 3 gives some clues in this direction but it would be good to see the background levels in Supp. Fig. 4a.

Response: We have added a line showing entry into the HEKdelta SIA cells to show background levels of entry that probably represent SIA-independent entry. The text was modified to describe this (line 179 to 181): "*Entry of HU02 into HEKΔSia cells and HEKΔSia cells transfected with ST6Gal1 showed the same basal level of entry that potentially represents some SIA-independent entry (Supplementary Fig. 4).*".

10. Supp. Fig. 4: The X axis is labeled with "Virus dilution". Should this not be "Relative virus concentration"? Furthermore, some numbers on the X axis labeling are in superscript.

Response: Yes, we agree this is a better description. In the legend we have now indicated that a two-fold dilution series of virus concentrations was used.

11. Supp. Fig 5: The authors should consider adding the amounts transfected and lectin used within the graph, to make it easier for the reader.

Response: We have made the figure better readable by following the suggestions of the reviewer (supplementary Figure 6).

12. L. 37-39 "IAV infection is initiated by binding of hemagglutinin (HA) on the viral envelope to epithelial cell surface-exposed Sia receptors." This is true for most strains of IAV but not for all as bat IAV of the H17 and H18 subtypes do not use sialic acid for host cell entry. The authors should specify, e.g. by saying "Human and avian IAV infection..." or include a statement about bat IAV.

Response: The reviewer is right, we have adapted the sentence to include this point as follows (line 37): "*IAV infection by genotypes H1 to H16*".

REVIEWERS' COMMENTS

Reviewer #1 (Remarks to the Author):

The revision of the manuscript entitled „ Human-type acid receptors contribute to avian influenza A virus binding and entry by hetero-multivalent interactions“ by Liu and colleagues was performed very detailed and clearly. From my perspective, the authors improved the manuscript significantly by adding more data (supplementary data), expanding/changing Figure 6, and by changing the phrasing and axis labeling of all uncertainties. The statistical procedures are now also clearly explained in the manuscript. Further, the authors provided in a careful response letter literature and explanations to justify their method procedures. I therefore recommend the article for publication without further changes regarding my comments.

Reviewer #2 (Remarks to the Author):

The authors have responded thoughtfully to the reviewers' comments. Errors pointed out by the reviewers have been corrected and clarifications added where requested.

Reviewer #3 (Remarks to the Author):

The reviewers have addressed my points adequately.